

# The Jensen wind farm parameterization for the WRF and MPAS models

Yulong Ma[1], Cristina L. Archer[1], and Ahmadreza Vasel-Be-Hagh[2]

[1]Center for Research in Wind (CReW), University of Delaware, Newark, DE 19716 (USA)
[2]Department of Mechanical Engineering, Tennessee Technological University, Cookeville, TN 38505 (USA)

**Correspondence:** Cristina L. Archer (carcher@udel.edu)

**Abstract.** Wind farm power production is known to be significantly affected by turbine wakes. When mesoscale numerical models are used to predict power production, the turbine wakes cannot be resolved directly because they are sub-grid features and therefore their effects need to be parameterized. Here we propose a new wind farm parameterization that is based on the Jensen model, a well-known analytical wake model that predicts the expansion and wind speed of an ideal wake. The Jensen parameterization is implemented and inserted into two commonly-used atmospheric numerical models: the Weather Research and Forecasting (WRF) Model and the Model for Prediction Across Scales (MPAS). In addition, the internal variability in wind speed and direction within a wind farm, the wind direction uncertainty, and the superposition of multiple wakes are taken into account with an innovative approach. The proposed approach and parameterization are tested against observational data at two offshore wind farms: Lillgrund (small in size and tightly spaced) and Anholt (large and widely spaced). Results indicate that power production is predicted more accurately with the Jensen than with the Fitch wind farm parameterization, which is the only one available in WRF. Power predictions with the Jensen parameterization are similar in WRF and MPAS. The sensitivity to grid resolution is small and the bias is generally low and negative. In conclusion, we recommend that the Jensen wind farm parameterization be used in the WRF and MPAS models, especially for coarse resolution, high turbine density, and wind directions aligned with the turbine columns.

## 1 Introduction

As modern wind farms increase in quantity and size, and as wind turbines expand in diameter, understanding their aerodynamic wakes becomes more critical. Wind turbine wakes significantly decrease the wind farm power production (Archer et al., 2018). Barthelmie et al. (2009) report that, in large offshore wind farms, wake losses are 10 to 20% of the total power output. These power losses are even more significant for wind farms with tightly-spaced turbines (Dahlberg, 2009). Therefore, despite the progress made in understanding wakes and wake losses (Fleming et al., 2019; Archer and Vasel-Be-Hagh, 2019; Simley et al., 2020; Stevens and Meneveau, 2020; Johlas et al., 2020; Zong and Porté-Agel, 2020; Nouri et al., 2020; Wu and Archer, 2021),





an improved understanding of wakes and a more accurate modeling of their impacts are still of significant practical interest for predicting wind farm power production, developing optimal layouts and control strategies, and quantifying the potential

unintended impacts of wind farms on the surrounding environment.

Due to the increase in computational power in recent years, it has been possible to study turbine wakes with numerical simulations. Mainly two numerical approaches are employed: large-eddy simulation (LES) and mesoscale modeling. High-resolution LES is the most accurate because it solves the fine-scale details of the wakes around the turbines at a grid resolution of 10 m or less (Lu and Porté-Agel, 2011; Archer et al., 2013; Xie and Archer, 2015; Wu and Porté-Agel, 2015; Xie and Archer,

2017). However, LES is highly computationally expensive and is conducted in an idealized environment by prescribing the inflow characteristics, which prohibit its applications to non-idealized, real-world simulations that span over multiple days or over large wind farms.

Mesoscale modeling, on the other hand, is much less computationally demanding and is often applied to real-world cases where the two-way interaction between the atmospheric boundary layer (ABL) and the wind farms is taken into account. It

therefore has more practical applications, such as estimation of the annual energy production of a wind farm or prediction of temperature changes caused by the wakes. In mesoscale numerical simulations, usually with horizontal resolution of the order of kilometers and vertical resolution of the order of tens of meters within the ABL, a wind turbine is often parameterized as an elevated momentum sink and a source of turbulence within the vertical levels of the turbine rotor disk (Fitch et al., 2012; Volker et al., 2015; Abkar and Porté-Agel, 2015; Pan and Archer, 2018).

One of the most widely used wind farm parameterizations is the Fitch scheme (Fitch et al., 2012) in the Weather Research and Forecast (WRF) model (Skamarock et al., 2008), referred to as WRF-Fitch hereafter, which has been used widely to investigate the wakes of wind farms and their impacts (Volker et al., 2012; Fitch et al., 2013a, b; Byrkjedal et al., 2014; Fitch, 2015; Jiménez et al., 2015; Eriksson et al., 2015, 2017; Pryor et al., 2018; Pan et al., 2018; Lundquist et al., 2019). In two recent studies, Pryor et al. (2019) and Shepherd et al. (2020) conducted high-resolution mesoscale simulations to analyze

the performance of WRF-Fitch in modeling the downstream wake effects and impact of wind turbine arrays on near-surface conditions. Xia et al. (2019) used WRF-Fitch to understand the underlying mechanisms of wind farm induced changes in near-surface temperature over west central Texas. Lee and Lundquist (2017) evaluated the performance of the WRF-Fitch in various atmospheric conditions for a wind farm of 200 1.5-MW wind turbines in central Iowa. They reported that meteorological conditions and vertical grid resolution significantly affect the performance of the model. We note that a recent paper by Archer

et al. (2020) identified two issues with WRF-Fitch (a code bug in the way the Fitch parameterization is inserted in the WRF and the excessive value of a coefficient used to calculate the turbine-added turbulence) that likely affected past studies that used WRF v.6 and higher.

Despite its wide adoption and use, WRF-Fitch has been found to generally underestimate wake losses and overpredict the power output of wind farms, especially when the wind is aligned with the turbine columns (Jiménez et al., 2012; Pan and

Archer, 2018). As discussed extensively in Pan and Archer (2018), the underlying issue with WRF-Fitch is that, just like most other wind farm parameterizations that have been proposed in the literature, it treats all the turbines that are positioned in the same grid cell in the same way. The individual turbine coordinates are just used to assign each turbine to the center of a grid



cell, effectively neglecting the turbine layout within each grid cell. Hence, all turbines in a grid cell are subject to the same inflow wind speed (i.e., spatially and ensemble-averaged velocity in the grid cell), regardless of their position, and their wakes, which are sub-grid features, are neglected. Eriksson et al. (2015) suggested increasing the horizontal grid resolution to 333 m or finer to improve the wake results with WRF-Fitch, which is unfeasible for large-scale or long-term simulations (Rai et al., 2019). Abkar and Porté-Agel (2015) attempted to account for the wind farm layout by introducing a correction parameter $\xi$. This parameter $\xi$, however, is sensitive to the wind farm density and wind farm layout (e.g., aligned vs. staggered) and has to be obtained from ad-hoc LES results.

To explicitly take into account the layout of the wind farm, here we propose to incorporate an analytical wake loss model into the wind farm parameterization. Analytical wake loss models are simplified representations of the wakes that are based on analytical equations for the wind speed deficit (Archer et al., 2018). The only prior study that incorporated an explicit expression for the wake deficit of each wind turbine was Pan and Archer (2018)'s hybrid parameterization, which was based on the geometric model by Ghaisas and Archer (2016). Although the hybrid parameterization performed very well when coupled with the WRF model, the applicability of the geometric model to any wind farm and any wind turbine is questionable, as it was calibrated based on one specific wind farm (Lillgrund in Sweden) and one wind turbine (Siemens 2.3 MW).

In this study we develop and apply a new wind farm parameterization, based on the Jensen model (Jensen, 1983; Katic et al., 1986), also known as Park model (Peña et al., 2014), to two mesoscale models: the WRF model and the Model for Prediction Across Scales (MPAS; Skamarock et al. 2012). The Jensen model was selected for this parameterization because it is possibly the most widely used analytical wake loss model (Katic et al., 1986; Kirchner-Bossi and Porté-Agel, 2018; Staid et al., 2018; Rivas et al., 2009; Vasel-Be-Hagh and Archer, 2017) and because it performs reasonably well regardless of the wind turbine layout or wind direction (Gaumond et al., 2014; Keane et al., 2016; Tian et al., 2017; Archer et al., 2018; Ritter et al., 2017; Ge et al., 2019). In particular, Archer et al. (2018) evaluated the performance of six popular analytical wake loss models – namely Jensen, Larsen (Larsen, 1988), Frandsen (Frandsen et al., 2006), two Gaussian models (Xie and Archer, 2015; Bastankhah and Porté-Agel, 2014), and the geometric model (Ghaisas and Archer, 2016) – using field data collected at three utility-scale wind farms: Lillgrund in Sweden, which is a mid-sized, closely-spaced, offshore wind farm with a regular layout; Nørrekær in Denmark, which is a small, moderately-spaced, inland wind farm with a regular layout; and Anholt in Denmark, which is a large, widely-spaced, offshore wind farm with an irregular layout. Every analytical wake loss model's performance varied from farm to farm and even wind direction to wind direction. Jensen stood out for its consistently strong performance and for rarely ranking last for all directions and all farms. The Jensen model assumes a top-hat distribution of the velocity deficit in every turbine wake and then applies superposition methods to account for the interaction among multiple wakes.

Like all other analytical wake loss models, the Jensen model was developed based on one implicit common assumption: that the upstream undisturbed wind speed and direction are the same for all turbines within the wind farm. With an increase in the size of modern wind farms, however, significant variability in the distribution of wind speed and direction within a wind farm is expected, due to surface heterogeneity and mesoscale weather systems (Van Der Laan et al., 2017; Peña et al., 2018). Neglecting horizontal variability within large wind farms could introduce inaccuracies into the annual energy production and power density predictions. This study addresses this issue and accounts for the variability of wind speed and direction using





an innovative approach. Additionally, four wake superposition methods, including one proposed here for the first time, are examined in combination with the new Jensen wind farm parameterization.

The impacts of turbines on the flow are still parameterized as an elevated momentum sink and a turbulence source, like those in WRF-Fitch. This study validates the Jensen parameterization's performance against observational data collected at two offshore commercial wind farms: Lillgrund (small in size and tightly spaced) and Anholt (large and widely spaced).

## 2    Framework of the wind farm parameterization in WRF and MPAS

### 2.1    The Fitch wind farm parameterization

In the Fitch parameterization, wind turbines are represented as elevated drag elements that reduce the wind speed and produce turbulent kinetic energy (TKE) at each vertical level $k$ that intersects the rotor. The momentum sink and TKE source terms induced by the turbines in a grid cell are proportional to the fractional rotor area contained in that level ($A_k$) and to the grid-cell horizontal wind speed at that level ($U_k$):

$$\frac{\partial u_k}{\partial t} = -\frac{1}{2}\frac{N_t}{A_{cell}}\frac{A_k C_T U_k u_k}{(z_{k+1}-z_k)}, \tag{1}$$


$$\frac{\partial v_k}{\partial t} = -\frac{1}{2}\frac{N_t}{A_{cell}}\frac{A_k C_T U_k v_k}{(z_{k+1}-z_k)}, \tag{2}$$

$$\frac{\partial TKE_k}{\partial t} = \frac{1}{2}\frac{N_t}{A_{cell}}\frac{A_k C_{TKE} U_k^3}{(z_{k+1}-z_k)}. \tag{3}$$

The power $P$ generated by the turbines in a grid cell is estimated as:

$$P = \frac{1}{2}N_t \rho A C_P U_h^3. \tag{4}$$

In these equations, $u_k$ and $v_k$ are the horizontal wind components, $z_k$ is the height of vertical level $k$, $N_t$ is the number of turbines in the grid cell, $A_{cell}$ is the horizontal cross-sectional area of the grid cell (note that $A_{cell}$ is calculated differently in WRF and MPAS with different grid shapes), $\rho$ is the air density (set to a constant, 1.23 kg m$^{-3}$), $A$ is the turbine rotor area (equal to $\frac{\pi}{4}D^2$, where $D$ is the rotor diameter), $U_h$ is the hub-height wind speed that is interpolated from the horizontal wind

speed at the vertical levels surrounding the hub height, $C_P$ and $C_T$ are the power and thrust coefficients (prescribed functions of $U_h$ and dependent on the turbine model), and $C_{TKE} = C_T - C_P$ is the so-called TKE coefficient. We note that $C_{TKE}$ as defined in the Fitch parameterization results in an overestimate of TKE (Abkar and Porté-Agel, 2015; Pan and Archer, 2018); Archer et al. (2020) proposed that $C_{TKE}$ should be revised to one quarter of the original value. This revised $C_{TKE}$ has been added in WRF v4.2.1 and is used in this study.





The turbine-induced TKE term from Eq. 3 is directly added to the Mellor–Yamada–Nakanishi–Niino Level 2.5 (MYNN)
Planetary Boundary Layer (PBL) scheme (Nakanishi and Niino, 2009) in the WRF model, as WRF-Fitch only works in combi-
nation with this particular PBL scheme. The Fitch wind farm parameterization has not been implemented in the MPAS model
v7.0 (the latest version). We inserted it in MPAS in this study. It should be noted that the added TKE is improperly advected
in the WRF model with the original Fitch parameterization, due to a code bug in versions v3.6 to v4.2, but it is fixed in v4.2.1
(Archer et al., 2020). We use the latest version of the models WRF v4.2.1 and MPAS v7.0 (with a modification in the added
TKE advection) in this study.

## 2.2   The Jensen wind farm parameterization

The Jensen wind farm parameterization consists of two steps: the estimation of the wind speed deficit of a single turbine wake
(via the Jensen wake model) and the wake superposition method to account for the interaction and overlapping of multiple
wakes.

### 2.2.1   The Jensen wake model

The normalized wind speed deficit $\delta$ is defined as:

$$\delta(x) = \frac{\Delta U}{U_\infty} = \frac{U_\infty - U(x)}{U_\infty},$$ (5)

where $\Delta U$ is the wind speed deficit, $U_\infty$ is the undisturbed hub-height wind speed, and $U$ is the (reduced) wind speed in the
wake that is a function of the downwind distance $x$ from the turbine. The Jensen wake model assumes a top-hat distribution
of the wind speed deficit in the wake, meaning that the (reduced) wind speed in the wake is assumed to be uniform along
$y$ and along $z$ within the edges of the wake itself for each downstream distance $x$ (i.e., the wake is three-dimensional but
axisymmetric). The hub-height wind speed at turbine $i$ caused by the wake of turbine $j$ is expressed as:

$$U_{ij} = U_\infty(1 - \delta_{ij}) = U_\infty \left[ 1 - \frac{2a}{\left(1 + 2k_w \frac{x_{ij}}{D}\right)^2} \right],$$ (6)

where $x_{ij}$ is the along-wind distance between turbines $i$ and $j$, $k_w$ is the rate of wake expansion (equal to 0.075 and 0.04 for
onshore and offshore wind farms, respectively, Archer et al. (2018)), and the induction factor $a$ is related to the thrust coefficient
$C_T$ by:

$$a = \frac{1 - \sqrt{1 - C_t}}{2}.$$ (7)

If turbine $i$ with rotor area $A_i$ is not perfectly aligned with the upstream turbine $j$ along the wind direction, the Jensen model
needs a modification to account for the fact that only a portion of its rotor ($A_{0,ij}$) is affected by the wake of turbine $j$, while
the rest of the rotor ($A_i - A_{0,ij}$) experiences the undisturbed wind speed $U_\infty$ (Archer et al., 2018):

$$U_{ij} = U_\infty \frac{A_i - A_{0,ij}}{A_i} + U_\infty(1 - \delta_{ij})\frac{A_{0,ij}}{A_i} = U_\infty - \delta_{ij}U_\infty \frac{A_{0,ij}}{A_i}.$$ (8)



The wind speed provided by Eq. 8 is effectively a rotor-average wind speed (also known as rotor-equivalent or rotor-layer wind speed), which is a better representation of the wind speed experienced by the entire rotor than the wind speed at the exact location of the hub, and ultimately provides a better estimate of the power production of the turbine (Choukulkar et al., 2016; St. Pé et al., 2018).

Lastly, the wake predicted by the Jensen model is conical and three-dimensional, but, in its original formulation, the only directions that matter are $x$, along which the wind speed decreases linearly in the wake, and $y$, along which the wind speed deficit is constant within the lateral wake edges and zero outside of them. This two-dimensional horizontal "slice" of the wake is then repeated at all vertical levels within the wake cone. In addition to being axisymmetric, the Jensen model was originally formulated for a shearless wind flow, in which the undisturbed wind speed did not vary with height but was equal to $U_\infty$ at all levels. However, a shearless flow hardly ever occurs in reality. To properly account for vertical wind shear, the wake can no longer be treated as axisymmetric. Here the wind speed deficit as a function of $z$ is obtained as follows. The undisturbed hub-height wind speed $U_\infty$ is used to calculate the hub-height (reduced) wind speed in the wake $U(x)$ and then the ratio $U(x)/U_\infty$ is used to multiply the wind speed at each vertical level $k$ within the cone cross section. By doing so, the proposed Jensen model is effectively three-dimensional and accounts for vertical wind shear, thus it can be perfectly integrated in WRF.

### 2.2.2 The wake superposition methods

When multiple wakes from multiple turbines $j$ ($j = 1...N$) overlap at turbine $i$, the incoming flow speed for turbine $i$ is calculated by a wake superposition method. A review of different methods is given in Porté-Agel et al. (2020). The two most common superposition methods of wind speed deficits are a linear superposition (Lissaman, 1979), called method M1 hereafter:

$$M1 : U_i = U_\infty - \sum_{j=1}^{N} \left[ \delta_{ij} U_\infty \frac{A_{0,ij}}{A_i} \right], \tag{9}$$

and a squared superposition (Katic et al., 1986), called method M2 hereafter:

$$M2 : U_i = U_\infty - \sqrt{\sum_{j=1}^{N} \left[ \delta_{ij}^2 U_\infty^2 \left( \frac{A_{0,ij}}{A_i} \right)^2 \right]}, \tag{10}$$

At the upwind turbine $j$, the incoming flow speed $U_j$ is not necessarily equal to the undisturbed wind speed $U_\infty$, since turbine $j$ itself may also be affected by wakes. As such, an alternative method (Voutsinas et al., 1990), named M3 hereafter, can be used to estimate $U_i$ based on a slight modification of Eq. 10:

$$M3 : U_i = U_\infty - \sqrt{\sum_{j=1}^{N} \left[ \delta_{ij}^2 U_j^2 \left( \frac{A_{0,ij}}{A_i} \right)^2 \right]}. \tag{11}$$

We propose a new wake overlap method, named M4, based on the superposition of not the wind speed deficits but the resulting wind speeds as follows:

$$M4 : U_i = \sqrt{\frac{\sum_{j=1}^{N} U_{ij}^2}{N}}, \tag{12}$$





where $U_{ij}$ is given by Eq. 8.

Note that these four methods are not derived from fundamental principles and therefore none of them conserves kinetic energy or momentum. Methods that are based on the linear or squared superposition of the wind speed deficits, like M1–M3, may, at least in principle, cause negative wind speeds because each additional wake further reduces the resulting wind speed. The advantages of method M4 are that it will never cause negative wind speeds and that it is well suited for overlapping wakes that come from different grid cells with different upstream wind speeds and wind directions. On the other hand, method M4 may underestimate the deficit in the case of perfectly aligned turbines because it tends to dilute the wakes of the nearest turbines with the partially-recovered wakes of those further upstream. While every wake superposition strategy introduces some level of under- or over-estimation, comparison with observational data, discussed in the next sections, indicates that M4 is generally the most accurate.

In the Fitch parameterization, there is no wake overlapping because each turbine in a grid cell is treated the same, i.e., as a front-row turbine with the same incoming flow speed $U_\infty$, and because there are no wakes at all inside the grid cell. Thus in Eqs. 1 through 4 the tendency and power at a wind turbine are simply multiplied by the number of turbines in the grid cell $N_t$ to give the overall contributions of the turbines. But in the Jensen parameterization, a slight modification is needed because each wind turbine in a grid cell is affected by the upstream wakes differently. Specifically, the wind speed at each vertical level that intersects the rotor ($U_k$) is not the same at all turbines, thus $U_k$ in Eqs. 1 to 3 is replaced by $U_i/U_h \times U_k$ as follows:

$$\frac{\partial u_k}{\partial t} = -\frac{1}{2}\sum_{i=1}^{N_t}\frac{1}{A_{cell}}\frac{A_k C_T\left(\frac{U_i}{U_h}U_k\right)u_k}{(z_{k+1}-z_k)}, \tag{13}$$

$$\frac{\partial u_k}{\partial t} = -\frac{1}{2}\sum_{i=1}^{N_t}\frac{1}{A_{cell}}\frac{A_k C_T\left(\frac{U_i}{U_h}U_k\right)v_k}{(z_{k+1}-z_k)}, \tag{14}$$

$$\frac{\partial TKE_k}{\partial t} = \frac{1}{2}\sum_{i=1}^{N_t}\frac{1}{A_{cell}}\frac{A_k C_{TKE}\left(\frac{U_i}{U_h}U_k\right)^3}{(z_{k+1}-z_k)}, \tag{15}$$

and $U_h$ in Eq. 4 is replaced by the rotor-equivalent wind speed $U_i$:

$$P = \frac{1}{2}\sum_{i=1}^{N_t}\rho A C_P U_i^3. \tag{16}$$

Note that the thrust and power coefficients are evaluated using the local $U_i$ and that the grid-cell averaged hub-height wind speed $U_h$ coincides with $U_{\infty,i}$. In principle, if the wind turbines in a grid cell are not affected by any upstream turbine wakes, the Jensen wind farm parameterization predicts exactly the same result as the Fitch parameterization.

### 2.2.3 Treatment of wind speed variability

In their original formulations, methods M1 to M4 are all based on the assumption that the undisturbed wind speed is uniform for all turbines in the entire wind farm and thus the equations for M1–M4 (Eqs. 9–12) contain only one value of $U_\infty$. When





they are implemented in a mesoscale numerical model, such as WRF and MPAS, the grid-cell horizontal wind speed is used as the undisturbed speed $U_\infty$. If the wind farm of interest is so large that it cannot be entirely contained in a single grid cell of the numerical domain, or if the grid resolution is fine, the undisturbed speed $U_\infty$ is replaced with $U_{\infty,i}$ in the first term and with $U_{\infty,j}$ in the second term of Eqs. 9–12, which are the grid-cell wind speed at the grid cell of turbine $i$ and $j$ at the beginning of the time step. Thus, the multi-cell versions of M1 to M4 are expressed as:

$$M1: U_i = U_{\infty,i} - \sum_{j=1}^{N} \left[ \delta_{ij} U_{\infty,j} \frac{A_{0,ij}}{A_i} \right], \tag{17}$$

$$M2: U_i = U_{\infty,i} - \sqrt{\sum_{j=1}^{N} \left[ \delta_{ij}^2 U_{\infty,j}^2 \left( \frac{A_{0,ij}}{A_i} \right)^2 \right]}, \tag{18}$$

$$M3: U_i = U_{\infty,i} - \sqrt{\sum_{j=1}^{N} \left[ \delta_{ij}^2 U_j^2 \left( \frac{A_{0,ij}}{A_i} \right)^2 \right]}, \tag{19}$$

$$M4: U_i = \sqrt{\frac{\sum_{j=1}^{N} \left[ U_{\infty,i} \frac{A_i - A_{0,ij}}{A_i} + U_{\infty,j} (1 - \delta_{ij}) \frac{A_{0,ij}}{A_i} \right]^2}{N}}. \tag{20}$$

As discussed later, all wake superposition methods M1–M4 are inserted in WRF and MPAS, and their performances are compared. Note that the multi-cell wake superposition methods also address the issue of wind speed variability within the wind farm because they directly use the wind speed at each grid cell. Also, Eqs. 13–16 do not need any modifications in multi-cell cases.

### 2.2.4 Treatment of wind direction variability and uncertainty

When turbines in a wind farm are located in multiple grid cells, the variability in wind direction within the wind farm needs to be taken into account (note that the variability in wind speed was addressed in the previous section with the multi-cell wake superposition methods). Here we propose an innovative approach that considers each individual turbine and all the relevant upstream turbines that might affect it. As displayed in Fig. 1, each grid cell within the wind farm contains one or more wind turbines and is affected by an upstream wind that may come from a different direction than that in the grid cell itself. To enhance computational speed, only the subset of turbines contained within an angle of $\pm 30°$ around the wind direction at the turbine of interest (red circle) and within a distance $<20\,D$ are considered (the turbines within the red dashed sector). The wind speed deficit caused by these upstream turbines is calculated along the direction that is the average (green arrow) between the wind vector at the upstream cell (black arrow) and that at the cell of the turbine of interest (red arrow). The calculations are conducted starting from the most upstream turbine and going downstream using the hub-height grid-cell wind components.

The wind direction defines the path of a turbine wake, which determines the wake conditions (e.g., full versus partial wake) at the downwind turbines. However, wind direction uncertainty exists in both measurements and numerical models (Gaumond



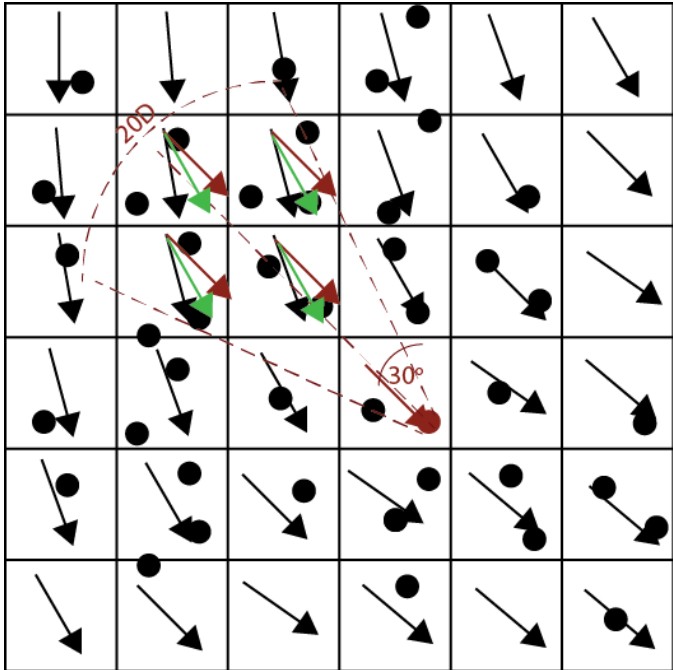

**Figure 1.** Example of how the wind direction variability within the farm is treated in the Jensen wind farm parameterization. The wind turbines are black circles and the turbine of interest is red. The green arrows are the average of the wind direction at the grid cells and that at the turbine of interest.

et al., 2014). Wind vanes and sonic anemometers have a wind direction accuracy that is typically $\pm 3°$ and $\pm 2°$ respectively
(Archer et al., 2016). This means that an observation of wind direction $\alpha$ is as likely to be from any direction between $\alpha$-3°
and $\alpha$+3° if a vane was used. For a mesoscale simulation, the timestep is roughly 10 s and the wind speed and direction are
fixed during this timestep. The wind at a timestep is essentially regarded as spatially- and ensemble-averaged within a grid cell.
In reality, many factors affect the flow causing uncertainty in the wind direction in this timestep (e.g., turbulent fluctuations,
turbine wake meandering, sub-grid phenomena, and uncertainty in the numerical model).

To address the wind direction uncertainty, following Gaumond et al. (2014) and Göçmen et al. (2016), a Gaussian wind
direction averaging method is used with a standard deviation of 2°. We choose seven angles around the flow direction $\pm 2.5°$,
$\pm 1.5°$, $\pm 0.5°$, 0° to apply the Gaussian weighted averaging, implying that the Jensen wind farm parameterization is called
seven times in a timestep, once for each of these seven wind directions. As such, the simulation results are effectively averaged
over a window of wind directions that is $\pm 2.5°$ around the intended wind direction. The output power, momentum sink, and
added TKE are all averaged using Gaussian weighted averages. This Gaussian wind direction averaging is not added in the
Fitch parameterization because it is insensitive to wind direction. We note that Volker et al. (2015) used nine wind directions
in a much wider range ($\pm 11.25°$) and ran nine cases with these wind directions separately. The observations in this study are
averaged over a $\pm 2.5°$ window too, but with no Gaussian averaging.



## 3 Methods and data

### 3.1 Wind power observations at two operational wind farms

The Jensen and Fitch parameterizations are evaluated against data collected at two offshore wind farms: Lillgrund, small in size and tightly spaced with a regular layout (Fig. 2a) and Anholt, large and widely spaced with an irregular layout (Fig. 2b). The Lillgrund wind farm is located in Sweden and consists of 48 Siemens 2.3-MW wind turbines with rotor diameter $D = 93$ m and hub height $H = 65$ m, for a total installed capacity of approximately 110 MW. The Anholt wind farm is located between

Djursland and the island of Anholt in Denmark and consists of 111 Siemens 3.6-MW wind turbines with $D = 120$ m and $H = 82$ m, for a total installed capacity of approximately 400 MW. The measurements at Lillgrund were approximately 16 months at a frequency of 1 min or less and those at Anholt were from January 2013 to June 2015 at a frequency of 10 min. No atmospheric stability filter is applied on the measurements, similar to the procedure in Volker et al. (2015). The observational datasets used here are the same as those in Archer et al. (2018); thus the readers are referred to that article for a detailed description of the

data cleaning procedure, including the yaw bias corrections. As a result of the data cleaning process, observations are only available for selected columns along and around a few directions of alignment at both farms, displayed in Fig. 2. We also analyzed data for two directions of non-alignment for each farm, shown in red in Fig. 2, selected because they were relatively close to directions of alignment (i.e., within $\pm 20°$).

### 3.2 Simulation setup in WRF and MPAS

In the simulations, we use the latest version of the models: WRF v4.2.1 and MPAS v7.0. To focus on the wind farm parameterization, we perform a series of idealized simulations under neutral stability conditions. Thus, for the physical options (note that MPAS uses the same physics packages from WRF), only the surface layer scheme and the PBL scheme are turned on. The surface layer scheme `sf_sfclay_physics` is set to 5, which provides the momentum drag over a sea surface, with the roughness length given by the Charnock relation (Charnock, 1955). The PBL scheme is MYNN level 2.5 model

(`bl_pbl_physics = 5`; Nakanishi and Niino 2009), which is the only available option with the Fitch wind farm parameterization.

     The simulations are carried out in a rectangular domain of 144 km × 144 km × 10 km in the $x$, $y$, and $z$ directions, respectively. The vertical resolution is 4.4 m near the surface and stretched to a 211-m grid spacing at the domain top, with a total of 62 vertical levels (of which four levels are below the rotor and nine levels intersect the rotor). To investigate the

sensitivity to the horizontal resolution, we set up three horizontal resolutions of $\Delta x = 4$ km (for Lillgrund, 24 km for Anholt, as explained later), $\Delta x = 2$ km, and $\Delta x = 1$ km, but keep the domain size (i.e., the length and width of the domain) the same. The bottom surface is set as water, with surface roughness calculated in the surface layer scheme. A rigid boundary is used at the top of the domain by setting the vertical velocity to zero. A Rayleigh damping layer is applied on the vertical velocity within the top 2000 m of the domain to prevent reflection of gravity waves. Open radiative lateral boundary conditions are used in

WRF, and periodic lateral boundary conditions are used in MPAS since the open boundary conditions are not available. Based



**Figure 2.** Wind turbine layout at Lillgrund (a) and Anholt (b). The Lillgrund wind farm is also plotted in (b) with red dots using the same scale as that for the Anholt wind farm. The black dashed lines indicate the wind directions (black arrows) aligned with wind turbine columns. The red arrows indicate the directions of non-alignment. The red dashed lines are the columns analyzed in this study for the non-alignment directions.

on our tests, as the wind farm wake extends less than 50 km downwind, the horizontal domain of 144 km × 144 km in MPAS is large enough to minimize the effect of the reentering of the wind farm wake.



The flow in the domain is driven by a pressure gradient that is balanced by a constant and uniform geostrophic wind, with the Coriolis parameter set to $1.11 \times 10^{-4}$ s$^{-1}$ at a latitude of $50°$N. With the imposed geostrophic wind (i.e., $u_g = 7.6$ m s$^{-1}$,

$v_g = 5.88$ m s$^{-1}$), the steady-state flow has a wind speed of about 8.5 m s$^{-1}$ at 70 m (i.e., about the hub height) from the wind direction $225°$. The flow in the whole domain is initialized with a constant potential temperature profile of 300 K from the surface to a height of 900 m and then linearly increasing with height at a constant rate of 3 K km$^{-1}$. The horizontal wind components are set equal to the geostrophic wind at all levels. The humidity is set to zero. All runs are integrated first for 96 hours without the turbines to reach a steady state. Then another four hours are run with the wind farm placed at the southwest

corner of the domain (i.e., wind farm center at one-third of the domain from the south and the west boundaries).

To ensure a proper comparison of the Jensen versus Fitch parameterizations in WRF and MPAS, the physical schemes and the domain configurations are set equal in the two models to the extent possible. Because the MPAS model uses hexagonal grids, here we define the horizontal grid resolution as the distance between the center of two adjacent grids. Due to the different grid shapes in WRF and MPAS (rectangle vs. hexagon), turbines may be partitioned into different grid cells even with the same

grid resolution. We also note that the turbine density ($N_t/A_{cell}$) in WRF and MPAS is different because of the difference in the grid shape (the area of a hexagonal grid is about 15% smaller than that of a rectangular grid with the same grid resolution). Even with the same configurations and for the same wind farm parameterization, the predicted power generation at the same wind farm is expected to be slightly different between the two models.

When the flow in the domain reaches a steady state after 96 hours, the wind farm is activated. Based on the wind farm

layouts, we choose six wind directions for Lillgrund (i.e., $180°$, $222°$, $255°$, $270°$, $300°$, and $315°$) and eight wind directions for Anholt (i.e., $168°$, $179°$, $183°$, $210°$, $228°$, $240°$, $339°$, and $341°$) to run the simulations. Here we rotate the wind farm layout and use the same steady flow with a direction of $225°$ for all runs, rather than performing the simulations initialized with different geostrophic winds to obtain the steady flow with the desired wind directions. The simulations are categorized into two groups based on the grid resolution: single- and multi-cell cases. For the single-cell cases, the entire wind farm is contained

in a single grid cell, while for the multi-cell cases, the turbines of the wind farm are located in multiple grid cells. For the single-cell cases, we use different grid resolutions at Lillgrund and Anholt, 4 and 24 km, respectively, because of the different wind farm sizes. For the multi-cell cases, we use two grid resolutions, 2 and 1 km, at both farms. As mentioned in section 22.2.4, to account for the wind direction uncertainty, the simulations with the Jensen parameterization and the observations are averaged over a wind direction sector of $\pm 2.5°$, or $5°$ wide.

**4 Results and discussion**

**4.1 Wake superposition methods**

Since one key factor in the Jensen parameterization is the superposition of multiple wakes, we first examine the four wake superposition methods, M1–M4, for the single-cell cases at Lillgrund and Anholt and then we choose the best two methods to apply in WRF and MPAS for multi-cell cases. For the single-cell cases, only the results from WRF are displayed because the

Jensen parameterization in WRF and MPAS produces identical results due to the same undisturbed wind speed and direction





used for all the turbines. For both the single- and multi-cell cases, power output from the Fitch parameterization is also shown for comparison.

Relative power is used for the comparative analysis, which is defined as the ratio of the power of each turbine in a column over that of the front-row turbine in the same column. The performance statistics used are bias error and root-mean-square
error (RMSE), defined as follows:

$$Bias = \frac{\sum_{i=1}^{N}\left(X_i - X_{i,obs}\right)}{N},\qquad(21)$$

$$RMSE = \sqrt{\frac{\sum_{i=1}^{N}\left(X_i - X_{i,obs}\right)^2}{N}},\qquad(22)$$

where $obs$ represents an observation, $X_i$ is the variable (i.e., relative power) of interest for turbine $i$, and $N$ is the sample size.
These statistics are expressed as a percent in this study.

At Lillgrund, we simulate the power output along eight columns for five directions of alignment (i.e., wind directions under full-wake conditions): 180°, 222°, 225°, 270°, and 300°; and along one column for one direction of non-alignment (partial wake condition): 315° (Fig. 2a). Note that, since Lillgrund is tightly spaced and regular, there is no direction that is truly of non-alignment; even 315° for the column led by turbine T46 is partially affected by wakes. For the wind directions of 180°,
222°, and 225°, two columns are selected because they represent two distinct situations in which one column includes a gap and the other does not.

At Anholt, we simulate the power output for six directions of alignment: 168°, 179°, 183°, 228°, 339°, and 341°; and two directions of non-alignment (no-wake condition): 210° and 240° (Fig. 2b). Because of the irregular wind farm layout, we choose the wind directions of alignment with as many turbines as possible aligned along them to run the simulations. Note that
the same directions of alignment and turbines were used in Archer et al. (2018).

### 4.1.1 Single-cell cases

Fig. 3 shows relative power from superposition methods M1–M4 in the Jensen parameterization at Lillgrund. The relative power from the Fitch parameterization is also shown for comparison, although it is always equal to one since the sub-grid turbine wakes are not considered.
For alignment and non-alignment directions, methods M1–M4 predict an identical and sudden drop in relative power at the second turbine. The identical relative power can be derived from Eqs. 9 to 12, which give the same expression when $N$ is equal to 1 for the single-cell version of the equations. However, this power drop at the second turbine is overestimated in all the columns compared with the observations, regardless of the distance between the first and the second turbine. A similar trend was also found in previous studies using the Jensen analytical wake loss model with a narrow wind averaging sector of
$\pm 2.5°$ (Simisiroglou et al., 2019). We note that, for the non-alignment wind direction 315° (Fig. 3i), methods M1–M4 predict an identical relative power at all turbines (except for the last one), because all but the last one experience only one upstream partial wake.



**Figure 3.** Lillgrund (single-cell): Relative power from observations and from the Jensen wind farm parameterization with the four wake superposition methods M1–M4 along the directions of alignment (a-h) and the direction of non-alignment (i). The black circles represent the mean observed relative power and the bars are one standard deviation. The simulation results from the Jensen parameterization and the observations are averaged over $\pm 2.5°$. The wind direction averaging was not applied to the results from the Fitch parameterization because it is insensitive to wind direction.





**Table 1.** Bias and root-mean-square error (RMSE) in power prediction by the Fitch parameterization and by the Jensen parameterization with the wake overlap methods M1, M2, M3, and M4 at Lillgrund for single-cell cases (in percent). The wind turbine columns are shown in Fig. 3a–h and the leading turbine of each column is indicated in parentheses where there is ambiguity. 'N.A.' in parentheses represents the non-alignment wind direction. The results are from the WRF model.

| | 180°(T23) | | 180°(T15) | | 222°(T23) | | 222°(T30) | | 255°(T36) | |
| Model | Bias | RMSE | Bias | RMSE | Bias | RMSE | Bias | RMSE | Bias | RMSE |
|---|---|---|---|---|---|---|---|---|---|---|
| Fitch | 51.2 | 56.1 | 47.9 | 53.0 | 56.5 | 60.5 | 51.4 | 56.0 | 44.2 | 49.5 |
| M1 | -21.3 | 25.5 | -23.1 | 26.6 | -21.6 | 24.7 | -21.9 | 25.4 | -21.9 | 26.3 |
| M2 | -16.7 | 18.8 | -15.4 | 17.2 | -18.1 | 19.7 | -17.7 | 19.6 | -15.3 | 17.2 |
| M3 | -7.0 | 7.8 | -7.0 | 8.0 | -6.4 | 7.4 | -7.7 | 9.1 | -7.7 | 9.9 |
| M4 | 2.6 | 8.9 | 1.5 | 9.1 | 4.7 | 10.5 | 0.6 | 8.7 | -0.7 | 10.1 |
| | 255°(T4) | | 270° | | 300° | | 315° (N.A.) | | All° | |
| Model | Bias | RMSE | Bias | RMSE | Bias | RMSE | Bias | RMSE | Bias | RMSE |
| Fitch | 42.8 | 49.3 | 38.7 | 45.7 | 54.9 | 58.8 | 10.7 | 12.9 | 44.3 | 49.1 |
| M1 | -19.6 | 22.8 | -13.5 | 15.9 | -26.4 | 29.6 | 2.6 | 6.1 | -18.5 | 22.5 |
| M2 | -10.5 | 11.8 | -6.3 | 10.4 | -22.3 | 24.6 | 3.1 | 5.6 | -13.2 | 16.1 |
| M3 | -4.0 | 7.1 | -1.6 | 11.4 | -13.4 | 14.5 | 3.2 | 5.6 | -5.7 | 9.0 |
| M4 | 2.6 | 8.9 | 1.5 | 9.1 | 4.7 | 10.5 | 0.6 | 8.7 | -0.7 | 10.1 |

After the second turbine, the four methods differ significantly. For most columns in Fig. 3, an increase in relative power at the third turbine, a phenomenon caused by the enhanced turbulence mixing after the first turbine that makes the second turbine wake recover faster, is reproduced well by M3 and M4, while M1 and M2 do not capture this feature. Not only at the third turbine, but generally, M1 and M2 (M1 in particular) substantially underpredict the power output at all the downwind turbines compared with the observations.

As shown in Table 1, M1 produces the largest overall bias (-18.5%) and the largest overall RMSE (22.5%), followed by M2 with a bias of -13.2% and RMSE of 16.1%. The reason why M1 and M2 perform worse is that they use the undisturbed wind speed in calculating the velocity deficit (the second term in Eqs. 9 and 10).

For the two better-performing methods M3 and M4, the performance of M3 is consistent at all the downwind turbines at Lillgrund, with most of its predictions lower than the observations, in agreement with Archer et al. (2018), but still within the error bars, whereas the performance of M4 is less consistent. As shown in Fig. 3, M4 tends to underpredict the power output at the near-front turbines (i.e., the second and third turbines) and overpredict the power output at the inner turbines (i.e., fourth, fifth, and following turbines). Additionally, M4 predicts the power increase caused by the gap well, while M3 still underpredicts the power there (Fig. 3b,d). The resulting overall bias for M4 is 2.5%, while that for M3 is -5.7% (Table 1). In terms of RMSE, the two superposition methods have close values, with a slightly larger overall RMSE for M4 (9.0% vs. 10.1%





for M3 vs. M4). In general, M3 and M4 perform similarly at Lillgrund for the single-cell cases, with M4 giving a slightly higher power prediction (slightly positive bias).

Fig. 4 presents comparisons of the relative power output from M1–M4 at Anholt. Although the turbine spacing and the turbine properties are different between the two wind farms, M1–M4 perform similarly as they did at Lillgrund. Again, M1 and M2 substantially underpredict the power at all the downwind turbines and perform worse than M3 and M4, with large overall RMSEs (23.0% and 15.5% for M1 and M2, respectively, Table 2). M3 tends to underpredict the relative power, while M4 slightly overpredicts it. Both M3 and M4 reproduce well the feature of power output becoming steady after the fourth

turbine in an alignment column (Fig. 4). The performance of M4, in particular the overprediction at inner turbines, seems to be improved with larger turbine spacing at Anholt.

For the non-alignment directions 210° and 240°, the Jensen and the Fitch parameterizations predict the same relative power (all equal to 1 except for the third turbine in the column for 240°). It is expected since the two parameterizations are essentially the same for the turbines that are not blocked by any upstream turbine wakes, as we pointed out in section 2.2.

In general, M4 slightly outperforms M3 at Anholt, with smaller overall RMSE and bias. As at Lillgrund, M4 tends to predict higher relative power than M3 also at Anholt, but still lower than the observed.

**Table 2.** As in Table 1 but at Anholt for single-cell cases (in percent). The results are from the WRF model.

| Model | 168° Bias | 168° RMSE | 179° Bias | 179° RMSE | 183° Bias | 183° RMSE | 228° Bias | 228° RMSE | 339° Bias | 339° RMSE |
|---|---|---|---|---|---|---|---|---|---|---|
| Fitch | 31.5 | 35.2 | 35.5 | 39.7 | 44.9 | 50.2 | 38.2 | 40.0 | 43.7 | 49.4 |
| M1 | -23.5 | 27.1 | -17.2 | 21.1 | -20.5 | 26.1 | -34.2 | 36.0 | -20.0 | 25.2 |
| M2 | -15.4 | 17.4 | -9.6 | 12.6 | -14.5 | 16.8 | -22.0 | 23.0 | -13.9 | 16.5 |
| M3 | -9.9 | 11.3 | -4.2 | 9.0 | -6.5 | 7.6 | -12.4 | 13.0 | -6.0 | 7.1 |
| M4 | -5.7 | 7.9 | -0.4 | 8.7 | 0.1 | 6.4 | -6.4 | 7.9 | -0.0 | 5.8 |

| Model | 341° Bias | 341° RMSE | 210° (N.A.) Bias | 210° (N.A.) RMSE | 240° (N.A.) Bias | 240° (N.A.) RMSE | All° Bias | All° RMSE | | |
|---|---|---|---|---|---|---|---|---|---|---|
| Fitch | 49.1 | 53.8 | 5.3 | 5.7 | 14.8 | 15.8 | 32.9 | 36.2 | | |
| M1 | -23.3 | 27.7 | 5.3 | 5.7 | 12.5 | 15.1 | -15.1 | 23.0 | | |
| M2 | -15.0 | 17.2 | 5.3 | 5.7 | 12.5 | 15.1 | -9.1 | 15.5 | | |
| M3 | -5.8 | 6.4 | 5.3 | 5.7 | 12.5 | 15.1 | -3.4 | 9.4 | | |
| M4 | 4.2 | 8.9 | 5.3 | 5.7 | 12.8 | 15.0 | 1.2 | 8.3 | | |

### 4.1.2   Multi-cell cases

The two better-performing methods M3 and M4 (multi-cell versions Eqs. 17 to 20) in the Jensen parameterization are further examined for multi-cell cases at Lillgrund and Anholt. For the multi-cell cases, the turbines in a wind farm are partitioned into

different grid cells, implying that the undisturbed wind speed might be different for each turbine (remember that we use the



**Figure 4.** Same as Fig. 3 but for Anholt (single-cell) with the directions of alignment (a-f) and the directions of non-alignment (g-h).





grid-cell hub-height wind speed as undisturbed wind speed for any turbine). Thus, differences in power output are expected due to the difference in grid resolutions and grid shapes (i.e., rectangle in WRF vs. hexagon in MPAS).

Figs. 5 and 6 show relative power from M3 and M4 at Lillgrund with grid resolutions of 2 and 1 km, respectively. The results from the Fitch parameterization are also shown to provide a comparison. Comparing the power output by the Jensen

parameterization in WRF and MPAS, it appears that the predicted power output from the two numerical models are fairly close, showing similar overall bias and RMSE at Lillgrund (Table 3). No general trend of under- or over-prediction by one model over the other was found. Small differences can be found between the two models because the wind turbines may be partitioned into different grid cells, depending on the grid resolution, grid shape, and wind direction (remember that we rotate the wind-farm layout to represent different wind directions).

The turbine partitioning affects the undisturbed wind speed in the grid cells, regardless of which wind farm parameterization is used, in two ways. First, the more the turbines partitioned into a specific grid cell, the larger the drag force exerted on the flow in that grid cell, resulting in a lower grid-cell wind speed at the end of each time step (which is the undisturbed wind speed for the turbines in that grid cell at the beginning of the next time step), and vice versa. Second, turbine partitioning affects the undisturbed wind speed via the numerical advection of lower wind speeds into downstream grid cells. As discussed later in

section 4.2, the combination of the decrease in wind speed due to the turbine drag and the numerical advection of this reduced wind speed is also called the "resolved wake" effect.

With respect to the performance of M3 and M4 in the Jensen parameterization at Lillgrund, we observe that M4 performs more consistently than M3 in the two numerical models (Figs. 5 and 6), suggesting that M4 is less sensitive to the turbine partitioning compared to M3. For example, at the second turbine in each column, the relative power predicted by Jensen with

M4 from WRF and from MPAS are nearly the same (although not identical), but the predictions by M3 differ markedly. When compared to the observations, in line with the single-cell cases, M4 always outperforms M3, particularly at finer grid resolutions. For the directions of non-alignment (Figs. 5i and 6i), more differences between the two numerical models and between the simulated and the observed values are found, especially at the last few turbines, due to the partitioning of the turbines into different grid cells. The reduction in relative power with grid resolution is attributable to the resolved wake effect,

which will be discussed in the next section.

Figs. 7 and 8 show relative power at Anholt with grid resolutions of 2 and 1 km, respectively. Besides the general trend discussed above (i.e., WRF and MPAS give very similar predictions and M4 predicts slightly higher relative power, which is closer to the observations than M3), the figures indicate little sensitivity of the relative power to grid resolution along directions of alignment at Anholt.

In summary, the Jensen wind farm parameterization with M4 still outperforms the others at Anholt, as indicated by the smallest overall bias and RMSE (Table 4). Also, increasing the resolution from 2 km to 1 km does not improve the power predictions at Anholt, as the bias and RMSE are about the same, or slightly worse.



Figure 5. Same as Fig. 3 but for Lillgrund multi-cell with $\Delta$x = 2 km.







**Figure 6.** Same as Fig. 3 but for Lillgrund multi-cell with $\Delta x = 1$ km.



**Table 3.** As in Table 1 but at Lillgrund for multi-cell cases (in percent). The results are from WRF and MPAS.

| Model | 180°(T23) Bias | RMSE | 180°(T15) Bias | RMSE | 222°(T23) Bias | RMSE | 222°(T30) Bias | RMSE | 255°(T36) Bias | RMSE |
|---|---|---|---|---|---|---|---|---|---|---|
| Δx = 2 km | | | | | | | | | | |
| Fitch-WRF | 42.4 | 47.9 | 35.4 | 41.1 | 43.0 | 48.3 | 34.8 | 39.9 | 26.9 | 32.9 |
| M3-WRF | -9.7 | 11.3 | -11.1 | 12.4 | -10.2 | 11.2 | -12.6 | 13.7 | -13.8 | 15.9 |
| M4-WRF | 2.3 | 8.3 | 0.8 | 8.3 | 3.7 | 8.9 | -1.4 | 7.7 | -1.8 | 8.8 |
| Δx = 1 km | | | | | | | | | | |
| Fitch-WRF | 24.0 | 31.0 | 20.0 | 30.0 | 21.4 | 25.7 | 6.9 | 13.1 | 10.0 | 17.6 |
| M3-WRF | -15.2 | 17.2 | -16.4 | 18.5 | -15.5 | 16.7 | -20.8 | 22.7 | -19.8 | 22.3 |
| M4-WRF | -0.9 | 5.1 | -2.8 | 6.5 | -2.7 | 5.3 | -9.8 | 11.3 | -5.6 | 8.6 |
| Δx = 2 km | | | | | | | | | | |
| Fitch-MPAS | 33.7 | 40.1 | 30.4 | 38.4 | 51.8 | 55.6 | 16.4 | 18.7 | 43.6 | 48.8 |
| M3-MPAS | -11.8 | 13.7 | -12.5 | 14.7 | -7.7 | 8.4 | -17.7 | 19.3 | -7.4 | 9.5 |
| M4-MPAS | 2.0 | 6.8 | 0.3 | 6.8 | 4.6 | 9.7 | -7.8 | 9.2 | 0.3 | 10.2 |
| Δx = 1 km | | | | | | | | | | |
| Fitch-MPAS | 13.2 | 17.6 | 5.6 | 15.9 | 24.2 | 30.4 | 23.3 | 33.8 | 8.1 | 10.3 |
| M3-MPAS | -17.2 | 19.3 | -20.7 | 23.2 | -15.4 | 16.7 | -16.4 | 18.4 | -20.0 | 22.8 |
| M4-MPAS | -5.1 | 6.6 | -8.0 | 9.1 | -2.3 | 5.2 | -3.8 | 8.4 | -8.0 | 10.1 |
| Model | 255°(T4) Bias | RMSE | 270° Bias | RMSE | 300° Bias | RMSE | 315° (N.A.) Bias | RMSE | All° Bias | RMSE |
| Δx = 2 km | | | | | | | | | | |
| Fitch-WRF | 16.4 | 18.6 | 19.7 | 24.0 | 40.1 | 43.9 | 4.4 | 8.1 | 29.2 | 33.9 |
| M3-WRF | -13.8 | 16.0 | -10.9 | 13.8 | -16.4 | 17.9 | -2.1 | 5.7 | -11.2 | 13.1 |
| M4-WRF | -2.9 | 7.3 | 1.8 | 13.6 | -3.0 | 6.6 | -1.1 | 4.9 | -0.2 | 8.3 |
| Δx = 1 km | | | | | | | | | | |
| Fitch-WRF | 7.3 | 8.5 | 14.9 | 20.9 | 22.2 | 30.5 | -16.4 | 22.8 | 12.3 | 22.2 |
| M3-WRF | -17.4 | 19.9 | -13.2 | 16.5 | -20.3 | 22.4 | -21.0 | 25.5 | -17.7 | 20.2 |
| M4-WRF | -5.5 | 8.5 | -0.3 | 11.9 | -5.8 | 6.9 | -19.1 | 24.4 | -5.8 | 9.8 |
| Δx = 2 km | | | | | | | | | | |
| Fitch-MPAS | 32.4 | 36.6 | 27.6 | 32.0 | 29.9 | 37.5 | -9.4 | 19.0 | 28.5 | 36.3 |
| M3-MPAS | -7.6 | 9.2 | -6.9 | 11.0 | -18.6 | 20.7 | -15.7 | 22.3 | -11.8 | 14.3 |
| M4-MPAS | 1.1 | 7.8 | 3.0 | 14.4 | -3.2 | 5.5 | -13.4 | 19.4 | -1.5 | 10.0 |
| Δx = 1 km | | | | | | | | | | |
| Fitch-MPAS | -1.8 | 3.7 | -0.3 | 0.7 | 13.9 | 19.1 | -23.1 | 25.2 | 7.0 | 17.4 |
| M3-MPAS | -21.5 | 24.5 | -20.4 | 24.7 | -21.3 | 23.0 | -27.2 | 29.6 | -20.0 | 22.5 |
| M4-MPAS | -7.7 | 10.2 | -5.2 | 10.3 | -8.3 | 9.4 | -25.6 | 27.8 | -8.2 | 10.8 |





**Table 4.** As in Table 1 but at Anholt for multi-cell cases (in percent). The results are from WRF and MPAS.

| Model | 168° Bias | 168° RMSE | 179° Bias | 179° RMSE | 183° Bias | 183° RMSE | 228° Bias | 228° RMSE | 339° Bias | 339° RMSE |
|---|---|---|---|---|---|---|---|---|---|---|
| Δx = 2 km | | | | | | | | | | |
| Fitch-WRF | 25.1 | 28.9 | 19.9 | 22.6 | 35.0 | 39.4 | 13.5 | 16.5 | 31.1 | 36.0 |
| M3-WRF | -13.5 | 15.3 | -12.7 | 14.8 | -10.1 | 11.4 | -23.0 | 24.3 | -11.0 | 12.6 |
| M4-WRF | -6.8 | 8.7 | -5.7 | 7.7 | -1.8 | 5.1 | -16.6 | 18.4 | -2.7 | 5.8 |
| Δx = 1 km | | | | | | | | | | |
| Fitch-WRF | 21.5 | 24.3 | 21.9 | 24.7 | 34.2 | 38.8 | 10.0 | 15.0 | 30.5 | 35.8 |
| M3-WRF | -14.6 | 16.3 | -10.4 | 13.4 | -10.1 | 11.4 | -24.1 | 25.7 | -11.3 | 12.9 |
| M4-WRF | -8.2 | 9.9 | -4.2 | 7.5 | -1.1 | 5.5 | -17.4 | 19.5 | -2.4 | 5.7 |
| Δx = 2 km | | | | | | | | | | |
| Fitch-MPAS | 18.3 | 21.7 | 19.3 | 22.2 | 36.2 | 41.1 | 20.0 | 22.4 | 36.0 | 41.8 |
| M3-MPAS | -16.8 | 18.9 | -13.0 | 14.9 | -9.9 | 11.4 | -19.8 | 21.0 | -9.3 | 10.9 |
| M4-MPAS | -8.9 | 10.5 | -5.5 | 7.6 | -1.3 | 5.1 | -13.5 | 15.1 | -1.4 | 5.4 |
| Δx = 1 km | | | | | | | | | | |
| Fitch-MPAS | 16.5 | 18.9 | 23.7 | 26.6 | 30.9 | 34.8 | 7.0 | 10.5 | 29.1 | 34.8 |
| M3-MPAS | -16.7 | 18.7 | -9.6 | 12.3 | -11.2 | 12.6 | -25.5 | 26.8 | -11.8 | 13.6 |
| M4-MPAS | -9.6 | 11.1 | -3.8 | 7.4 | -2.4 | 5.3 | -19.0 | 20.9 | -2.6 | 5.9 |

| Model | 341° Bias | 341° RMSE | 210° (N.A.) Bias | 210° (N.A.) RMSE | 240° (N.A.) Bias | 240° (N.A.) RMSE | All° Bias | All° RMSE |
|---|---|---|---|---|---|---|---|---|
| Δx = 2 km | | | | | | | | |
| Fitch-WRF | 41.7 | 46.3 | -9.6 | 12.2 | -12.2 | 14.7 | 18.1 | 27.1 |
| M3-WRF | -8.4 | 9.5 | -9.6 | 12.2 | -14.0 | 15.3 | -12.8 | 14.4 |
| M4-WRF | 3.4 | 7.5 | -9.6 | 12.3 | -14.0 | 15.2 | -6.7 | 10.1 |
| Δx = 1 km | | | | | | | | |
| Fitch-WRF | 41.4 | 46.0 | -6.9 | 9.5 | -16.7 | 19.0 | 17.0 | 26.6 |
| M3-WRF | -8.5 | 9.6 | -6.8 | 9.5 | -18.5 | 20.0 | -13.0 | 14.8 |
| M4-WRF | 3.5 | 8.0 | -6.9 | 9.5 | -17.8 | 19.3 | -6.8 | 10.6 |
| Δx = 2 km | | | | | | | | |
| Fitch-MPAS | 42.2 | 46.9 | -18.3 | 19.3 | -6.1 | 10.0 | 18.4 | 28.2 |
| M3-MPAS | -8.1 | 9.3 | -18.2 | 19.2 | -8.0 | 11.5 | -12.9 | 14.6 |
| M4-MPAS | 3.7 | 8.1 | -18.2 | 19.2 | -7.2 | 10.2 | -6.5 | 10.2 |
| Δx = 1 km | | | | | | | | |
| Fitch-MPAS | 29.9 | 33.4 | -14.5 | 15.6 | -15.6 | 17.9 | 13.4 | 24.1 |
| M3-MPAS | -12.1 | 13.3 | -14.4 | 15.5 | -17.3 | 18.8 | -14.8 | 16.4 |
| M4-MPAS | 0.3 | 4.6 | -14.4 | 15.5 | -16.7 | 18.2 | -8.5 | 11.1 |





**Figure 7.** Same as Fig. 3 but for Anholt multi-cell with $\Delta x = 2$ km.

## 4.2 Sensitivity to grid resolution

The combination of the wind speed reduction due to the extraction of momentum from the flow caused by each additional
turbine in a grid cell (i.e., wind turbine drag) and the numerical advection of this reduced wind speed from upstream into
downstream grid cells can be considered as the "resolved" wind farm wake (Jiménez et al., 2015; Eriksson et al., 2015). When



**Figure 8.** Same as Fig. 3 but for Anholt multi-cell with $\Delta x = 1$ km.

the Fitch parameterization is used, this is the only wake effect that is accounted for; when the Jensen parameterization is used, this effect is still present but it is masked or compensated for by the sub-grid wakes.

To examine the sensitivity of the resolved wakes to grid resolution, we first analyze the results from the Fitch parameteri-
zation, where the situation is less complicated without considering sub-grid wakes. In general, the power predictions with the



Fitch parameterization decrease as the grid resolution increases. This sensitivity can be observed in both relative power and total power, as discussed next.

The progressive decrease in relative power from the Fitch parameterization along long columns of turbines can be appreciated, for example, for the 222° case at Lillgrund (Fig. 6c) or the 228° at Anholt (Fig. 8d), none of which is manifest in the observations. Long columns of turbines are more likely to be partitioned over multiple grid cells; thus, the resolved wake effect is further amplified.

Comparing relative power from the Fitch parameterization at 2 km versus 1 km, either at Lillgrund (Figs. 5 and 6) or Anholt (Figs. 7 and 8), the values are generally lower at the finer resolution, for both alignment and non-alignment directions. For example, at 300° at Lillgrund, the relative power at the last turbine is ∼0.6 at 2 km but ∼0.4 at 1 km for WRF and MPAS; at Anholt, at 228° the relative power at the last turbine is >0.7 at 2 km but ∼0.6 at 1 km for WRF and MPAS. Again, this effect is more pronounced at long columns because of the repeated resolved wake effect (i.e., numerical advection and momentum extraction).

The power production of the entire wind farm (Fig. 9) also decreases as the grid resolution gets finer when the Fitch parameterization is used. Note that in Fig. 9 the undisturbed hub-height wind speed is the same for all wind directions and for all methods. Since the results from MPAS are similar to those from WRF, only the WRF results are shown, but the conclusions drawn are still valid for MPAS. The observational data are not displayed in this figure since they are not available for the entire wind farm, as mentioned in section 3.1. However, the power predictions for the selected columns for which data were available, shown in Fig. 10, will also be discussed later.

For all the wind directions at Lillgrund and Anholt, the power output in the single-cell cases is the largest with Fitch. This is consistent with the single-cell results from the previous sections, where the relative power was (incorrectly) always 100%. When further increasing the grid resolution to 2 or 1 km, notable differences are observed between the single-cell and multi-cell cases with Fitch, with power production being more than 15% lower at the finest resolution (multi-cell 1-km). This is because the resolved wake effect causes an incremental reduction of wind speed deep within the wind farm as the grid resolution increases. As shown in Fig. 9b, this trend (i.e., power output decreases with increasing grid resolution) is also evident at Anholt, although the magnitude of the decrease going from 2 to 1 km is smaller than at Lillgrund for two reasons. First, the turbine density ($N_t/A_{cell}$) is much lower at Anholt than at Lillgrund and, second, the turbine density at Anholt is about the same at 2 km and 1 km due to the wide spacing.

Despite the resolved wakes, the Fitch parameterization substantially overpredicts power along directions of alignment, particularly for the coarse resolution cases (Fig. 10), suggesting that considering only the resolved wakes might not be adequate. By contrast, for directions of non-alignment (e.g., 315° at Lillgrund and 210° and 240° at Anholt in Fig. 10), the power predictions with the Fitch parameterization are slightly underestimated, consistent with the literature (Jiménez et al., 2015; Pan and Archer, 2018). The underestimate for directions of non-alignment occurs because the resolved wakes incorrectly reduce the value of the hub-height wind speed used to calculate power downstream ("incorrectly" because there are no wake losses for non-alignment directions). Increasing the grid resolution is definitely beneficial with the Fitch parameterization, especially for



**Figure 9.** Total power (MW) at Lillgrund (a) and Anholt (b) from the Jensen wind farm parameterization in WRF with the two wake superposition methods M3 (red) and M4 (blue) and from the Fitch parameterization (grey) from all the turbines in the farm for the selected wind directions. For each method, single-cell cases (SC) and multi-cell cases with grid resolutions of 2 km and 1 km are shown. The figures with 'N.A' in the title are the results for the non-alignment wind directions (315° at Lillgrund, 210° and 240° at Anholt).





directions of alignment, because the reduction in wind speed and power caused by the resolved wakes is amplified with finer resolution.

**Figure 10.** Same as Fig. 9, but for selected directions at Lillgrund and Anholt for which observations were available.



The Jensen parameterization includes the sub-grid wake effects, which are less sensitive to grid resolution and tend to decrease the total power output when compared to Fitch's predictions. In particular at Lillgrund, where the sub-grid wake effects are strong, the wind farm power predicted by the Jensen parameterization with M3 and M4 is much smaller than that by
Fitch (Fig. 9). Both M3 and M4 predict power outputs that are much less sensitive to grid resolution than Fitch, especially when the turbine density is high, like at Lillgrund. M4 always predicts a higher power output than M3, closer to the observations (Fig. 10); this trend is more pronounced at Lillgrund with strong sub-grid wakes.

Note that, for directions of non-alignment, both Fitch and Jensen perform well at both farms (Fig. 10) and the predictions do not improve with finer grid resolution, not even for Fitch.

Unlike in Fitch's, the resolved wake effect is small in Jensen's due to the much larger effect of the sub-grid wakes. Relative power decreases slightly as the grid resolution increases at Lillgrund (e.g., compare Figs. 5 and 6), due to the resolved wake effect. At Anholt, a small decrease in power output can be detected for all directions when the resolutions changes from 2 km to 1 km (Figs. 9b and 10b). This is the small "signature" of the resolved wake effect in Jensen's results.

At Lillgrund, because of the regular layout, the trends in total power as a function of grid resolution (Fig. 9a) are very similar
to those from the selected columns for each direction of alignment, for which observational data are available (Fig. 10a). At Anholt, however, since the wind farm layout is irregular, only a small fraction of the turbines are truly aligned with any given wind direction and therefore the displayed wind farm power output in Fig. 9b includes multiple wake conditions (i.e., full, partial, and no wakes; actually, most turbines are under partial or no wake conditions), resulting in a much higher wind farm efficiency. As such, some of the trends of wind farm output differ slightly from those along the columns with data. For example,
for the non-alignment direction 210°, the power in Fig. 10b is basically equal at 2 km and 1 km and between M3, M4, and Fitch, whereas it exhibits minor differences in Fig. 9b. Despite these minor inconsistencies, the grid sensitivity of the entire wind farm power is generally the same as that at the selected columns with observations at Anholt too.

We conclude that the Jensen parameterization, especially with M4, performs best at both farms. The Jensen parameterization tends to underestimate the power, regardless of the alignment or non-alignment conditions and regardless of the grid resolution.
The consistent sign of the bias (negative) in column power output and the absence of sensitivity to the grid resolution or wind direction are all desirable properties. By contrast, the Fitch parameterization is sensitive to the grid resolution and to the alignment or non-alignment of the turbine columns with the wind directions. For directions of alignment, Fitch's largely overestimates the power for single- and multi-cell cases at both farms. At Lillgrund, however, at 1 km, the power output predicted by the Fitch parameterization is close to the observed for wind directions of 255° and 270° (those with the largest
spacing, 6D and 8.5D respectively), but it is still generally overestimated for the other wind directions. For directions of non-alignment, the Fitch parameterization tends to slightly underestimate the power, more so at fine resolution and at Anholt (i.e., widely-spaced farm).

With respect to the wake overlapping methods with the Jensen parameterization, we recommend method M4, although M3 also performs satisfactorily for single-cell cases. We note that M4 consistently predicts higher power output than M3.





### 4.3 Wind speed and TKE distributions

To obtain a better understanding of how the two wind farm parameterizations affect the power production of a wind farm, we compare the simulated wind speed and TKE at Lillgrund and Anholt. Since the two parameterizations perform similarly in WRF and MPAS, we only show the results from WRF here. The wake superposition method in the Jensen parameterization is M4.

Fig. 11 shows the vertical profiles of wind speed and TKE from the Fitch and the Jensen parameterizations from a single-cell case at the grid cell containing the wind farm (Lillgrund or Anholt). The wind direction for the selected case at Lillgrund is $222°$ and that at Anholt is $168°$. For these wind directions, the wake effects are the strongest, which is favorable for comparison. Note that the grid resolution for the single-cell case at Lillgrund is 4 km, while that at Anholt is 24 km, resulting in a significant difference in the turbine density between two wind farms (i.e., $3.0/\mathrm{km}^2$ vs. $0.2/\mathrm{km}^2$).

Comparing the wind speed profiles from the two wind farm parameterizations, we find a much larger wind speed reduction from the Fitch parameterization at Lillgrund, but surprisingly, the difference between the two parameterizations is almost indistinguishable at Anholt. This different behaviour at Lillgrund and Anholt is attributed to the turbine density and the wind farm layout. At Lillgrund, the turbine spacing is small and the wind-farm layout is regular, leading to a large grid-cell averaged turbine drag force on the flow. At Anholt, the turbine spacing is large and the wind-farm layout is irregular, such that most of the turbines are not (or partially) influenced by sub-grid turbine wakes; thus, the Jensen parameterization behaves similarly to the Fitch parameterization. As a result, when the turbine-induced drag forces are averaged over a large grid cell of 24 km × 24 km at Anholt, the difference in the speed reduction between the two parameterizations is almost indistinguishable with respect to the background wind speed.

For the predicted TKE from the two parameterizations, however, differences are observed at both wind farms, with a larger TKE value from the Fitch parameterization. At Anholt, although the wind speeds from the two wind farm parameterizations are nearly the same (Fig. 11c), the TKE profiles are more notably different (Fig. 11d). For example, the turbine-added TKE at hub-height at Anholt is about $0.087 \mathrm{~m}^2 \mathrm{~s}^{-2}$ for the Jensen parameterization and $0.098 \mathrm{~m}^2 \mathrm{~s}^{-2}$ for the Fitch parameterization. Compared to the TKE background value of $0.48–0.51 \mathrm{~m}^2 \mathrm{~s}^{-2}$, the value of the turbine-induced TKE source is much larger than the value of the turbine-induced momentum sink, about 20% versus 1%, for both parameterizations.

We note that the TKE profile at Lillgrund is different from that in Pan and Archer, 2018 (their Fig. 8a) with the same grid resolution and hub-height wind speed. Our result from the Fitch parameterization is much lower (e.g., $0.9 \mathrm{~m}^2 \mathrm{~s}^{-2}$ vs. $1.55$ $\mathrm{m}^2 \mathrm{~s}^{-2}$ for the maximum TKE from the two) and the maximum TKE occurs near hub height, while their maximum value is located slightly above hub height. These differences are attributable to the fixed bug in the TKE advection in the WRF code (Archer et al., 2020). Along with the bug fix, we also followed Archer et al. (2020)'s suggestion to reduce the TKE coefficient $C_{TKE}$ to one quarter of the original value.

At both Lillgrund and Anholt (Figs. 12 and 13), the wind speed at the grid cell of the wind farm is slightly lower and the TKE is larger in the Fitch parameterization than in Jensen's, in line with the vertical profiles from the single-cell case. The

**Figure 11.** Vertical profiles of wind speed (a,c) and TKE (b,d) by the Fitch and Jensen parameterizations from WRF single-cell cases, 222°
at Lillgrund (upper) and 168° at Anholt (lower).

differences between the two parameterizations are more pronounced at Lillgrund, where the wake losses are more significant
due to the tighter spacing than at Anholt.





**Figure 12.** Horizontal cross sections of: (a,b) hub-height wind speed (m s$^{-1}$), (c,d) hub-height TKE (m$^2$ s$^{-2}$), and (e,f) power production (MW) at Lillgrund from the Jensen (left) and Fitch (right) parameterizations. The results are from WRF with grid resolution of 1 km and wind direction of 222°.





525     In general, even though a difference in the wind speed between the two wind farm parameterizations is observed, this difference is small (e.g., $\sim$0.1 m s$^{-1}$ at hub height for the single-cell case at Lillgrund). Thus, the large difference in the predicted power output between the two parameterizations is mainly determined by the consideration of sub-grid wakes. Our results indicate that the Jensen parameterization, which accounts for the sub-grid wakes, may tend to underpredict the power output in multi-cell cases.

## 530   5   Conclusions

We describe and implement a new wind farm parameterization in the WRF and MPAS mesoscale models, based on the classical Jensen model, which takes into account the sub-grid wind turbine wakes, the wind speed and direction variability within a wind farm, and the wind direction uncertainty. Four turbine wake superposition methods are examined in the Jensen parameterization, including an innovative method (i.e., M4) that is based on the superposition of the resulting wind speeds in the wake, as opposed to the resulting deficits. The observational data at two offshore wind farms, Lillgrund and Anholt, are used to evaluate the wind farm parameterization. The Jensen wind farm parameterization performs well in the WRF and MPAS models as the results of the simulations match the observations, particularly with the new proposed wake superposition method M4, while the Fitch wind farm parameterization tends to overpredict the power, especially at coarse grid resolution and for directions of alignment. The simulation results at the two wind farms with totally different grid spacing and layout show similar overall bias and RMSE, suggesting the robustness of the Jensen parameterization. Grid sensitivity tests show that, while the Fitch parameterization tends to predict decreasing relative power at downwind turbines and decreasing total power output with increased grid resolution, due to the resolved wakes, this effect is minimal in the Jensen results.

    The proposed Jensen wind farm parameterization predicts the power generation at each wind turbine, without increasing too much the computational cost (about 2%), which makes it a valuable tool for practical applications, such as estimation of the annual energy production and optimization of wind-farm layout. Future implementations may consider a more sophisticated formulation of the wake expansion coefficient $k_w$, using for example turbulence intensity and/or atmospheric stability (Stevens et al., 2016); a more advanced wake loss model, such as the Gaussian model by Xie and Archer (2015); and a separate model for the TKE in the wake (one that would not rely on advection alone to inject the correct amount of TKE in the wake of the wind farm), to improve the wind farm power and flow prediction in mesoscale numerical simulations.

550   *Code and data availability.*   The authors are in the process of making the Jensen code available through the WRF github website. The wind farm data are proprietary and may not be distributed.

*Author contributions.*   CLA and VB designed the study and obtained funding; YM coded the parameterization, ran the simulations, and lead the manuscript preparation. All authors contributed to the manuscript writing.



**Figure 13.** Same as Fig. 12, but at Anholt for wind direction of 168°. Note that the flow appears to be from the southeast, but in reality it is from 168° and the wind farm is rotated accordingly.



*Competing interests.* The authors declare no competing interests.

555    *Acknowledgements.* This research was funded by the U.S. Bureau of Ocean Energy Management (BOEM), project n. 0040420490. The simulations were conducted on the Caviness high-performance computer cluster of the University of Delaware.



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
