# Peer review of "The Jensen wind farm parameterization for the WRF and MPAS models"

_Wind Energy Science, 2022_

## Referee Comment (RC1)

In the Fitch scheme, as in other schemes, drag forces are applied in turbine containing grid-cells. Then, (I) the WRF model dynamics handles the interaction between horizontal grid-cells through advection, lowering the wind speed inside a wind farm as grid-cells encounter lower wind speeds from up-stream cells and (II) model physics determinates the downstream vertical expansion of the wind speed deficit through turbulent diffusion. In this way the wind speed will, according to the model dynamics/physics, decrease downstream in the wind farm until an equilibrium between the energy extracted and the energy supply from above is reached. This means that the WRF model determines the downstream development of the wind speed, also within wind farms. In this approach the Jensen method is used to estimate the downstream wind speed $U_i$ inside a wind farm that is then used to estimate a wind speed reduction $(U_i/U_h)$. Some of my concerns are that the calculation of $U_i$ is not consistent with the WRF wind field and that the use of $U_h$ does not follow the definition of a free stream wind speed. Considering, furthermore, the increasing size of modern wind turbines, the turbine density per grid-cell will eventually reach one. Methods, as the proposed one, trying to estimate sub-grid wind speeds would only introduce errors. In the case of one turbine per grid-cell $U_i/U_h$ should end up being one, which especially in *real* mode simulations) is not guaranteed at all.

**Major comments**:

(1) Initially, when introducing the Jensen model you start by using $U_\infty$ for the velocity deficit and at l.160 you determine a wind speed reduction multiplying the wind speed by $U(x)/U_\infty$. Later in eq. 13/14 the wind speed reduction is replaced by a multiplication of the wind speed by $U_i/U_h$, stating (l.201) that $U_h$ coincides with $U_{\infty,i}$. I doubt that $U_h$ can be seen as the free stream wind speed $U_\infty$. Here, free stream meaning without influence of the wind farm. One should be aware that the wind speed reduction due to the wind farm starts well ahead of it due to the induced positive pressure gradients. On the other hand a $U_h$ is the average wind speed of a turbine containing grid-cell, possibly in the middle of the wind farm. This wind speed is already influenced by all the turbines in that cell (and additionally by turbines in front of that cell). Therefore, in my opinion, $U_\infty$ can not be replaced by $U_h$. How to determine a free stream wind speed: A free stream wind speed could be determined for *idealized* simulations, but how would you determine a free stream wind speed in *real* mode simulations (or even if you simulate wind

farm clusters), with a strongly inhomogeneous wind field?

(2) *Section The wake superposition methods:*
(I) I could not find your eq. 10 in Katic et al. 86 that only uses velocity deficits.
(II) The methods M1, M2 and M3 all depend (through $\delta_{ij}$) on the downstream distance $x$. Instead, the proposed M4 method seems not to depend on a downstream distance $x$. If this is the case, I would expect it to give the same wind speeds for a narrowly and widely spaced turbines and is therefore not very useful to predict wind speeds inside wind farms.
(III) Regarding the M4 method. One would assume that proceeding downstream the wind speed would decrease and ideally reach at a certain point an equilibrium. When I tried just an example with 3 turbines having wind speeds of 7, 6 and 5m/s. Then, the wind speed of the 4th turbine would be sqrt(110/3) = 6.1m/s, which is above the 5m/s of the 3th turbine. However, it is also not clear if the M4 method would not just start at the first turbine and proceed one by one downstream. If the first turbine would face say 8m/s, then the wind speed at the second turbine would be $U_2 = \sqrt{64/1} = 8\,\mathrm{m/s}$ and at the thirst turbine it would be $U_3 = \sqrt{(64+64)/2} = 8\,\mathrm{m/s}$, is this right?

(3) You mention that the wind speeds in large wind farms are not expected to be homogeneous (l.87-l.93) and that it should be accounted for. This is exactly why physics parametrisations operate one dimensionally and do not intervene in the horizontal model direction. In this way the wind speed field remains the inhomogeneous model wind speed and a wind speed reduction in a turbine containing grid-cell follows the local grid-cell velocity. On the other hand, this approach seems to assume a constant wind speed in the downstream distance, when calculating downstream wind speeds $U_i$ throughout the whole wind farm (row) and neglects therefore WRFs wind speed variability in the downstream direction. Consequently, the calculated downstream wind speed $U_i$ seems not in line with WRFs wind field. Problems can be expected when (large) wind farms are in regions with large wind speed gradients (coastal wind farms) and in unsteady flows.

(4) In case of multiple upstream wind speeds the M4 method changes and becomes a function of the rotor area, instead of being dependent on wind

speeds only. Could this sudden change be clarified?

(5) The total thrust per mass for a turbine $i$ that faces a wind speed $U_i$ should be proportional to $U_i^2$. Eqs 13 and 14 look differently. Could you explain the difference?

(6) Figs.11 a and c show the wind speed. However, it is important to understand additionally the shape of the (normalised) wind speed deficit: at which height is the maximum deficit? Is there an acceleration at the lowest model level? Theory predicts for the far wake in neutral conditions a Gaussian shaped deficit with a maximum deficit on hub-height (confirmed by CFD and LES results). If a result show other features they should be discussed and compared to reference studies (neutral measurements/LES simulations). The (normalised) wind speed deficit could be obtained by $(u(z) - u_0(z))/u_{0,h}$, where $u_0$ is the free stream velocity. Volker et al. used e.g. a reference simulation without wind farms (same simulation time) to determine the free stream velocity.

(7) Fig. 4. You show point measurements indicated by dots (they shouldn't be connected with straight lines) and model results. My question for Anholt is, how did you plot the figures? Are the small dots above the turbine numbers the grid-cell centres? Also, because in the figure all the distances for the 1-86 row seem the same, whereas in the layout the distance between turbine 1 and 31 seems to be larger than the distance between the other turbines in that row.

(8) One other concern is the intension to compare model results inside wind farms with measurements. The measurements are locally in strongly inhomogeneous conditions and the model instead is by design very diffusive in the horizontal direction with a true resolution far less than the model grid-size. Therefore, one should be very careful when trying to compare those two worlds.

**Further comments**:

l.141: this link between the induction factor and thrust coefficient is only

when momentum theory is applied.

l.118: in the original definition the added TKE is the difference between the power extracted from the flow and the power converted in electricity. How, can the one quarter of the original value be theoretically justified?

l.18: it should be that wind turbine wakes *can* significantly decrease the wind farm power production. With high wind speeds and/or large turbine spacing the reduction at a downstream turbine can be low (or approach zero).

l.39: it should be written that Volker et al. 2015 does *not* apply a TKE source term. Instead, it calculates a grid-cell averaged deceleration and the WRF model calculates the TKE due to the changed wind shear.

l.187: as stated in comments before, I would not agree that the first turbine row would face $U_\infty$. Also, what do you exactly mean with: there are no wakes at all inside the grid-cell? There is a grid-cell averaged deceleration, which is the consequence of the wakes inside the grid-cell.

l.64: could you please clarify what do you mean with *ad-hoc* LES simulations.

l.105: the three equations look a bit misleading, since there is only one source term listed in the model's deceleration (the same holds for the TKE). It would be clearer to introduce an additional source to the WRF deceleration (e.q. force/mass) and define the magnitude of that.

l.185: what do you mean with: M4 is *generally* more accurate? and why?

---

## Referee Comment (RC2)

**Review of *The Jensen wind farm parameterization for the WRF and MPAS models* by Y. Ma et al.**

Reviewer: M. Paul van der Laan, DTU Wind Energy

May 31, 2022

The authors develop a meso-scale wind farm parametrization based on the Fitch scheme coupled with an engineering wake model to represent the internal wake losses. In addition, a new wake superposition method is proposed. The predicted internal wake losses are validated with field measurements.

Wind farm parametrizations cannot resolve internal wake losses and I think it is very interesting to overcome this by coupling an existing parametrization with an engineering wake model. The article has a focus on the internal wake losses; however, wind farm parametrizations in meso-scale models are mainly employed to estimate wind farm wake losses (for example wind farms situated in a wind farm cluster). Hence, I lack a validation of a wind farm wake produced by your proposed parametrization. You could for example look at the Horns Rev I wind farm wake as used by Volker et al. [1] or simply compare the wind farm wakes of the employed models (new and old) for the present idealized setup. More detailed comments are listed below; they need to be addressed before the article can be considered for publication in Wind Energy Science. Note that I cannot provide detailed comments on the numerical setup of the meso-scale models due to a lack of experience regarding this type of model. Also note that I have not yet looked at any other reviewer's comments before uploading this document because I want to provide an independent review.

**Main comments**

1. Line 26: You mention that mainly two numerical approaches are employed to study wind turbine wakes: LES and WRF. However, there are a lot more numerical approaches to simulate wind turbine interaction, each model has its application area and purpose. Ranking steady-state wake models from low to high model fidelity one could think of: engineering wake models, simplified Reynolds-averaged Navier-Stokes (RANS) models as 2D RANS, parabolic RANS models [Iungo et al. (2018) [2]], linearized RANS (FUGA); and full 3D RANS with actuator disks (AD)[3]. Then we have transient wake models: Dynamic wake meandering model, unsteady RANS with actuator lines, LES-AD, LES with actuator lines (AL), WRF-LES-AD and WRF with a wind farm parametrization. Many of these model are described in Göçmen et al. (2016) [4], which you already refer to.

2. Line 74: Here you mention that the Jensen model works well for any wind farm layout based on the work of Archer et al. (2018) [5]. I do not agree with this strong statement since the performance of the model heavily relies on the wake superposition method and wake expansion coefficient. For example, the Jensen model does not work well for a tightly packed wind farm as Lillgrund and a below-rated inflow wind speed when the original quadratic wake superposition is applied (M2 method in the present work) [6], as you also show in the results section. I could not find the chosen superposition method in the work of Archer et al. (2018) [5], which makes is difficult to understand the results in that work. I think it would be better to write that the Jensen model could be calibrated to get the desired result for a given wind turbine type and wind farm layout.

3. Section 2.2.2: It is interesting that you investigate different superposition methods. What is the physical meaning of method M4? I find it strange to superpose the absolute wind

speed instead of the wake deficit. It means that you take a L2 norm of the wake wind speed normalized by the number of upstream turbine wakes, which is a form of averaging rather than a summation or superposition method. In other words, method M4 will always smooth out the wake effects between the largest and the smallest (upstream) wake wind speeds. There are also situations where the superposition method does not make sense. For example if one has a regular layout with a large lateral spacing (in $y$) but a small stream-wise spacing (in $x$), such that the turbines do not interact laterally, then a downstream wind turbine, will still experience reduced wake effects from the lateral turbines because you divide by the number of upstream turbines $N$.

4. How is the freestream wind speed defined/ calculated in your model? If it is the cell average, then this would mean that you would need to iterate over the Jensen model, since the cell average influences the internal wake model results, which influence the cell average. In case you use the freestream from the inflow, which is possible for an idealized case, how would you extend this to a realistic meso-scale setup where the inflow is transient?

5. How do you calculate the cell average wind speed from the Jensen model? Do you use the cell area only (and thus you would disregard the part of the wakes that extend beyond the cell area) or do you use an area that covers the entire wake of all the turbines within a cell?

6. Equation (14): Should the left hand side be $\frac{\partial v_k}{\partial t}$?

7. Line 240: You propose to use a Gaussian filter with a standard deviation of $2°$, where did you base this value on?

8. Line 284: Why do you use a latitude of $50°$? Lillgrund and Anholt are located at $55.5°$and $56.6°$, respectively.

9. Section 3.1: Why are the measurements not filtered for neutral conditions? Without such a filter it is not fair compare the numerical results for neutral conditions with the measurements. If you prefer to keep the current measurements results you need to clearly state (in results section and conclusion) that the model validation is not entirely fair.

10. Section 3.2: It seems that you are modeling an idealized setup in the mesocale models. How do the steady-state inflow profiles of wind speed, wind direction and TKE/TI look like for both WRF and MPAS? You could plot these results in Fig. 11. What is the actual roughness length applied by the Charnock relation? What is the turbulence intensity at the investigated hub heights?

11. Figures 3-8, measurements: What is the wind direction bin size used for the observations? In addition, you could consider to plot the error bars as the uncertainty of the mean (standard deviation dived by the square root of the number of samples).

12. I do not think it is fair to normalize the wind turbine power of Fitch with the power of the first row wind turbine, as Fitch is meant to model the average power of a number of wind turbines (or entire wind farm for the single cell case). You could overcome this by normalizing the wind turbine power of all the models and measurements with a hypothetical free-standing wind turbine power (using the free-stream value at the wind farm location [without any wind turbines present] and the power curve. The measured free-stream could be evaluated by using the power of a group of front row turbines and the power curve. A similar approach was performed in Hansen et al. (2015) [7].

13. Line 478: You mention that the Jensen parametrization is insensitive to grid resolution: *The Jensen parameterization tends to underestimate the power, regardless of the alignment or non-alignment conditions and regardless of the grid resolution. The consistent sign of the bias (negative) in column power output and the absence of sensitivity to the grid resolution or wind direction are all desirable properties.* However, in order to show a grid independence you need to compare results of at least three different grid levels and show that the results converge with horizontal grid refinement. For example, you can add another grid level in Figs. 9 and 10, for example for 0.5 km or for 4 km. (The 4 km cell size might not be interesting for

Lillgrund as this is the same at the single cell approach. (I am aware that grid-independence is challenging for WRF due to parametrizations that rely on large horizontal cells, but it might be possible for an idealized setup.)

14. The cases considered in this article only look at high thrust coefficients, since a below rated inflow wind speed is used. The conclusion on the best superposition method might change if you had considered an above rated wind speed, where the thrust coefficients are smaller. This it because the performance of a super position method can be shown to be dependent on the thrust coefficient, see for example Machefaux et al. (2015) [8]. You could add an above-rated case or you could add a comment/discussion in the article.

15. Line 370: You write: *Both M3 and M4 reproduce well the feature of power output becoming steady after the fourth turbine in an alignment column (Fig. 4).* I understand what you mean, but I think it is better to write about a balance instead of a steady-state. Here, I mean a balance between the momentum extracted by the wind turbines and the momentum transport into the wind farm from the boundary layer.

16. Figure 11: I would normalize the wind speed by the freestream wind speed, $U_\infty$, and the TKE by $\sqrt{2/3\text{TKE}}/U_\infty$. The height can be normalized as $(z - z_H)/D$, with $z_H$ and $D$ as the hub height and rotor diameter, respectively. The same applies to Figs. 12 and 13, where the cell power production could be normalized by the total wind farm power.

17. Line 496: You mention that a wind direction of $222°$ results in the largest wake losses for Lillgrund but this should 120 or $300°$, where the wind turbine spacing is smallest.

18. Line 547: When referring to the Gaussian wake model, I think you need to refer to the original work of Bastankhah and Porté-Agel (2014) [9].

**Minor comments**

1. Thanks for referring to my work. My last name is written with small letters for the first two words: *van der Laan*.

**References**

[1] P. J. H. Volker, J. Badger, A. N. Hahmann, and S. Ott, "The explicit wake parametrisation v1.0: a wind farm parametrisation in the mesoscale model wrf," *Geoscientific Model Development*, vol. 8, no. 11, pp. 3715–3731, 2015. [Online]. Available: https://gmd.copernicus.org/articles/8/3715/2015/

[2] G. V. Iungo, V. Santhanagopalan, U. Ciri, F. Viola, L. Zhan, M. A. Rotea, and S. Leonardi, "Parabolic rans solver for low-computational-cost simulations of wind turbine wakes," *Wind Energy*, vol. 21, no. 3, pp. 184–197, 2018. [Online]. Available: https://onlinelibrary.wiley.com/doi/abs/10.1002/we.2154

[3] M. P. van der Laan, N. N. Sørensen, P.-E. Réthoré, J. Mann, M. C. Kelly, N. Troldborg, K. S. Hansen, and J. P. Murcia, "The $k$-$\varepsilon$-$f_P$ model applied to wind farms," *Wind Energy*, vol. 18, no. 12, pp. 2065–2084, dec 2015. [Online]. Available: https://onlinelibrary.wiley.com/doi/10.1002/we.1804

[4] T. Göçmen, M. P. van der Laan, P. E. Réthoré, A. Peña Diaz, G. C. Larsen, and S. Ott, "Wind turbine wake models developed at the technical university of Denmark: A review," *Renewable and Sustainable Energy Reviews*, vol. 60, p. 752, 2016. [Online]. Available: https://dx.doi.org/10.1016/j.rser.2016.01.113

[5] C. L. Archer, A. Vasel-Be-Hagh, C. Yan, S. Wu, Y. Pan, J. F. Brodie, and A. E. Maguire, "Review and evaluation of wake loss models for wind energy applications," *Applied Energy*, vol. 226, pp. 1187–1207, 2018. [Online]. Available: https://www.sciencedirect.com/science/article/pii/S030626191830802X

[6] PyWake development team, "Automated validation report for PyWake," DTU Wind Energy, Tech. Rep., 2022. [Online]. Available: https://topfarm.pages.windenergy.dtu.dk/PyWake/validation.html

[7] K. S. Hansen, P.-E. Réthoré, J. Palma, B. G. Hevia, J. Prospathopoulos, A. Peña, S. Ott, G. Schepers, A. Palomares, M. P. van der Laan, and P. Volker, "Simulation of wake effects between two wind farms," *Journal of Physics: Conference Series*, vol. 625, p. 012008, jun 2015. [Online]. Available: https://doi.org/10.1088/1742-6596/625/1/012008

[8] E. Machefaux, G. C. Larsen, and J. P. M. Leon, "Engineering models for merging wakes in wind farm optimization applications," *Journal of Physics: Conference Series*, vol. 625, p. 012037, jun 2015. [Online]. Available: https://doi.org/10.1088/1742-6596/625/1/012037

[9] M. Bastankhah and F. Porté-Agel, "A new analytical model for wind-turbine wakes," *Renewable Energy*, vol. 70, pp. 116–123, 2014, special issue on aerodynamics of offshore wind energy systems and wakes. [Online]. Available: https://www.sciencedirect.com/science/article/pii/S0960148114000317

---

## Author Comment (AC1)

**Response to Reviewer #1 – Patrick Volker**

In the Fitch scheme, as in other schemes, drag forces are applied in turbine containing grid-cells. Then, (I) the WRF model dynamics handles the interaction between horizontal grid-cells through advection, lowering the wind speed inside a wind farm as grid-cells encounter lower wind speeds from up-stream cells and (II) model physics determinates the downstream vertical expansion of the wind speed deficit through turbulent diffusion.

In this way the wind speed will, according to the model dynamics/physics, decrease downstream in the wind farm until an equilibrium between the energy extracted and the energy supply from above is reached. This means that the WRF model determines the downstream development of the wind speed, also within wind farms. In this approach the Jensen method is used to estimate the downstream wind speed Ui inside a wind farm that is then used to estimate a wind speed reduction (Ui/Uh).

Some of my concerns are that the calculation of Ui is not consistent with the WRF wind field and that the use of Uh does not follow the definition of a free stream wind speed.

> *The argument that Ui may be inconsistent with the WRF wind field arises from a misunderstanding of the way the Jensen parameterization works and will be addressed at issue (3) on p. 5 of this response.*
> *The issue that Uh is not an obvious proxy for the free stream wind speed is valid in principle and we address it extensively at issue (1) on p. 2. In brief, the advantages offered are greater than the inconsistency and the alternatives are more arbitrary and prone to errors.*

Considering, furthermore, the increasing size of modern wind turbines, the turbine density per grid-cell will eventually reach one. Methods, as the proposed one, trying to estimate sub-grid wind speeds would only introduce errors. In the case of one turbine per grid-cell Ui/Uh should end up being one, which especially in *real* mode simulations) is not guaranteed at all.

> *While we agree that in future wind farms it is likely that the wind turbine density will reach one turbine per grid cell, most of the wind farms today are still going to be around in the next 20-25 years and will need to be properly parameterized in future WRF simulations for wind energy and weather forecasting applications. On the other hand, the grid resolution in WRF should not be finer than 1-2 km (Lean and Clark 2003, Lynn and Yair 2010, Kanth and Rao 2011, Calmet et al. 2018, among others). Thus, wind farms like Lillgrund and Anholt will still require more than one wind turbine per grid cell. Thus a wind farm parameterization that accounts for sub-grid wakes, like ours, will still be needed for a long while.*
> *Even with one turbine per grid cell, especially for cases of alignment or partial waking, our Jensen parameterization is very likely to outperform the Fitch's because, as we further explain below, relying on the resolved wake alone is not sufficient due to the diffusive nature of the numerical model.*
> *Ui/Uh should not be 1 except in three cases: at the front-row turbines, at turbines that are unaffected by wakes, and at distances such that the upstream wake has fully*

*recovered (after 20D in our code). Jensen will properly capture this even with one turbine per grid cell and even with empty cells in between because, although the upstream turbines might be in other grid cells, their impact on the wind speed that a turbine in a downstream grid cell experiences is accounted for. Hence, Ui will not be equal to Uh unless the turbine of interest is in one of the three cases mentioned above. By contrast, with Fitch's, Ui =Uh always, which is incorrect.*

**Major comments**:
(1) Initially, when introducing the Jensen model you start by using $U\infty$ for the velocity deficit and at l.160 you determine a wind speed reduction multiplying the wind speed by $U(x)/U\infty$. Later in eq.13/14 the wind speed reduction is replaced by a multiplication of the wind speed by $U_i/U_h$, stating (l.201) that Uh coincides with $U\infty,i$. I doubt that Uh can be seen as the free stream wind speed $U\infty$. Here, free stream meaning without influence of the wind farm. One should be aware that the wind speed reduction due to the wind farm starts well ahead of it due to the induced positive pressure gradients. On the other hand a Uh is the average wind speed of a turbine containing grid-cell, possibly in the middle of the wind farm. This wind speed is already influenced by all the turbines in that cell (and additionally by turbines in front of that cell). Therefore, in my opinion, $U\infty$ can not be replaced by Uh. How to determine a free stream wind speed: A free stream wind speed could be determined for *idealized* simulations, but how would you determine a free stream wind speed in *real* mode simulations (or even if you simulate wind farm clusters), with a strongly inhomogeneous wind field?

*The argument that Uh does not exactly follow the definition of free upstream wind speed is valid. We struggled with it ourselves, but eventually we concluded that any alternative would be even more arbitrary and would potentially introduce more errors than using the grid cell wind speed. For example, we could try to identify an upstream grid cell that could be considered free stream. But this grid cell would be wind direction dependent and grid size dependent. If you have a wind farm that is distributed among dozens of grid cells, each with its own wind direction and wind speed, which wind direction do you even pick to identify an upstream grid cell? If the wind farm is, say, 50 km long, does it really make sense to pick an upstream wind speed (and direction) that is 60 km away? What about the possible induction zone? We concluded that any other choice would be more arbitrary than just choosing Uh.*
*In addition, a parameterization should be easy to implement and fast. It would be impractical to propose a parameterization that relies on idealized simulations without the farm. Choosing Uh is straightforward, easy to implement, and appears to work well.*

*We changed the notation at lines 160 to be consistent and replaced U(x)/U_inf with Ui/U_inf.*

*We also added this discussion on p. 8:*
*"Alternative choices could be made for Uinf,i, such as the wind speed at a grid cell located at some distance upstream of the wind farm along one wind direction (among the varying wind directions simulated inside the farm), or the wind speed from offline*

*simulations without the wind farm. However, any alternative would be even more arbitrary and would potentially introduce more errors than simply using the local grid-cell Uh. Another advantage is that Uh is straightforward to implement in the codes and fast to execute."*

(2) Section The wake superposition methods:
(I) I could not find your eq.10 in Katic et al. 86 that only uses velocity deficits.

*The equation given in the Katic et al. (1986) paper is:*

$$\left(1 - \frac{V}{U}\right)^2 = \left(1 - \frac{V_1}{U}\right)^2 + \left(1 - \frac{V_2}{U}\right)^2$$

*where U is the free stream velocity. Each term in the parentheses is a normalized wind speed deficit, thus, using our notation, this equation can be rewritten as:*

$$\delta^2 = {\delta_1}^2 + {\delta_2}^2 = \sum_{i=1}^{2} \delta_i^2$$

*which is the sum of the squared deficits as stated in our manuscript.*

(II) The methods M1, M2 and M3 all depend (through $\delta_{ij}$) on the downstream distance x. Instead, the proposed M4 method seems not to depend on a downstream distance x. If this is the case, I would expect it to give the same wind speeds for a narrowly and widely spaced turbines and is therefore not very useful to predict wind speeds inside wind farms.

*Method M4 is a function of x, via the U_ij term (given by Eq. 8), which is a function of $\delta_{ij}$.*

(III) Regarding the M4 method. One would assume that proceeding downstream the wind speed would decrease and ideally reach at a certain point an equilibrium. When I tried just an example with 3 turbines having wind speeds of 7, 6 and 5m/s. Then, the wind speed of the 4th turbine would be sqrt(110/3) = 6.1m/s, which is above the 5m/s of the 3th turbine. However, it is also not clear if the M4 method would not just start at the first turbine and proceed one by one downstream. If the first turbine would face say 8m/s, then the wind speed at the second turbine would be U2 = sqrt(64/1)=8 m/s and at the thirst turbine it would be U3 = sqrt((64+64)/2) = 8 m/s, is this right?

*The reviewer is correct that M4 will always give you basically an average of the wind speeds of all the wakes involved, which by definition will be higher than the lowest wind speed (and lower than the highest). This makes most sense when the turbines are actually **not** aligned with the wind direction, which is the case for which this method was designed (multi-cell).*
*To understand this, let's look at the figure below, in which each turbine is in its own grid cell and the wind directions are slightly different in each grid cell, as is the case in real simulations (also the wind speeds are different, of course).*

[Figure]

*Turbine 4 is affected by 3 wakes, each of which causes its own wind speed (e.g., V4,1 is caused by turbine 1 and is a partial wake case, with some of the rotor affected by the wake and the rest unaffected, thus experiencing Vh,4). The example that you suggested is such that V4,2 = 7 m/s, V4,1 = 6 m/s, and V4,3 = 5 m/s. These are the resulting wind speeds at turbine 4, they are not the wind speeds at the upstream turbines. [Maybe this was a reason of confusion?] The resulting wind speed at turbine 4 will be 6.1 m/s, as you calculated, as a result of the mixing of the three individual wakes. Why should the resulting wind speed be lower than 5 m/s? If that was the case, neither mass nor momentum nor energy would be conserved and the equilibrium that the reviewer correctly referred to would never occur (it would monotonically decrease).*

*If, as the reviewer implicitly assumed, the turbines are aligned, then this is less intuitive and may even appear to be incorrect. We are used to seeing the squared deficits from the Jensen's top hat shape overlapping and reducing the wind speed more and more with more overlapping wakes via the M2 method. This is actually not happening in the real world. We know that the wind power and wind speed plateau very quickly at the second or third turbine in a row, due to partial wake recovery and entrainment. If you look at the figures in our manuscript for directions 222° or 180° at Lillgrund, or 339° and 183° at Anholt, you will see observational data that indicate that the third turbine generates more power than the second. With method M2 this cannot be reproduced, but with M4 it is.*

*Method M2 (and M3) does not conserve energy or mass or momentum. Why should the squared deficits be conserved? They are a proxy for kinetic energy, but the mass is not*

*accounted for and air is incompressible, thus density cannot change and it is physically impossible that kinetic energy sums up that way. However, the research community has been doing it regardless and has suffered the consequences, i.e., more significant errors than those of M4.*

*To shed more light on the underlying physics of M4 versus M2, let us assume n equally-distant aligned turbines. Then the equation for M2 is basically a geometric series like this (where x is the normalized deficit):*

$$\sum x^{2n}$$

*with |x|<1. Thus, as n increases, it converges to:*

$$\lim_{n \to \infty} \sum x^{2n} = \frac{1}{1-x} > 1$$

*which is greater than 1. This means that each additional wake deficit incrementally reduces wind speed and can even cause a zero wind speed or even negative. This is not physical and would never occur in reality. Method M4, however, does not suffer from this problem.*

*M4 does not conserve mass, momentum, or energy either, but it will never result in a negative or zero wind speed and can be thought of a way to account for entrainment and partial wake recovery. With M4, the resulting wind speed will be the average of the (partially recovered and therefore relatively high) wind speed from the farthest turbine and that (fresh and relatively low) of the nearest turbine, plus all the others in between. M4 effectively avoids the unrealistic continuous drop in wind speed as more and more turbines are aligned that other wake superposition methods suffer from. The fact that it outperforms other methods is also an indication that it is not incorrect.*

*We added the following additional sentence on p. 7:*
*"In a sense, M4 can be thought of as a way to indirectly account for partial wake recovery due to the added turbulence from multiple wakes and entrainment."*

(3) You mention that the wind speeds in large wind farms are not expected to be homogeneous (l.87-l.93) and that it should be accounted for. This is exactly why physics parametrisations operate one dimensionally and do not intervene in the horizontal model direction. In this way the wind speed field remains the inhomogeneous model wind speed and a wind speed reduction in a turbine containing grid-cell follows the local grid-cell velocity. On the other hand, this approach seems to assume a constant wind speed in the downstream distance, when calculating downstream wind speeds Ui throughout the whole wind farm (row) and neglects therefore WRFs wind speed variability in the downstream direction. Consequently, the calculated downstream wind speed Ui seems not in line with WRFs wind field. Problems can be expected when (large) wind farms are in regions with large wind speed gradients (coastal wind farms) and in unsteady flows.

*We do not assume a constant wind speed along the downstream distance. There is a different Uh in each grid cell and a different Ui at each turbine in each grid cell. We were very careful with our notation and only used i,j for the turbines, not for the grid cells. Thus in a grid cell with cell-average hub-height wind speed Uh, there are several values of Ui, one for each turbine within that grid cell, calculated via the Jensen model. The next grid cell will have another value of Uh and so on. The wind speed variability in a multi-cell farm is fully retained.*

(4) In case of multiple upstream wind speeds the M4 method changes and becomes a function of the rotor area, instead of being dependent on wind speeds only. Could this sudden change be clarified?

*The M4 method does not change, it was a function of the rotor area before as well (compare the expression for Ui given by Eq. 8 with that given in Eq. 20). The only difference is that for the single-cell case there is only one value of U_inf, whereas in the multi-cell case there are different values of U_inf, one from each relevant grid cell, due to the wind direction variability.*

(5) The total thrust per mass for a turbine i that faces a wind speed Ui should be proportional to $U_i^2$. Eqs 13 and 14 look differently. Could you explain the difference?

*Each turbine i experiences a wind speed Ui at hub height. In Eq. 13 and 14 we instead want the change in the wind velocity components u_k and v_k at each level k. We first calculate the wind speed at that level U_k, and then we reduce it by an amount that is proportional to the reduction at hub height Ui/Uh. This new speed is then multiplied by u_k or v_k to get units of m2/s2.*
*Eqs. 13 and 14 are the same as Fitch's Eqs. 1 and 2, but they are using a reduced U_k (due to the sub-grid wakes, via the correction factor Ui/U_h) rather than the original U_k. Basically, in both the Fitch and Jensen parameterizations, the thrust force is calculated at each level using the wind speed at that level (U_k), not the hub-height wind speed (Ui). At hub height, however, Uk=Uh and the equation becomes the familiar one.*

(6) Figs.11 a and c show the wind speed. However, it is important to understand additionally the shape of the (normalised) wind speed deficit: at which height is the maximum deficit? Is there an acceleration at the lowest model level? Theory predicts for the far wake in neutral conditions a Gaussian shaped deficit with a maximum deficit on hub-height (confirmed by CFD and LES results). If a result show other features they should be discussed and compared to reference studies (neutral measurements/LES simulations). The (normalised) wind speed deficit could be obtained by (u(z)−u0(z))/u0,h, where u0 is the free stream velocity. Volker et al. used e.g. a reference simulation without wind farms (same simulation time) to determine the free stream velocity.

*This paper should not be treated as a study of fundamentals of wind turbine wakes, rather as an improvement over the existing default wind farm parameterization by Fitch*

*et al. (2012). Your request to run additional simulations without wind farms in order to calculate a normalized wind speed deficit is extremely difficult for us to do at this point due to a lack of resources. Comparing Jensen results side-by-side with Fitch's indicates that the two are extremely consistent with one another, e.g., the wind speed deficit peaks at hub height for both. However, please let us know if something is suspicious and the results of those additional simulations would help clarify. We will try to obtain the resources required for executing them.*

(7) Fig. 4. You show point measurements indicated by dots (they shouldn't be connected with straight lines) and model results. My question for Anholt is, how did you plot the figures? Are the small dots above the turbine numbers the grid-cell centres? Also, because in the figure all the distances for the 1-86 row seem the same, whereas in the layout the distance between turbine 1 and 31 seems to be larger than the distance between the other turbines in that row.

*The distances between turbines in the figure are not proportional to the real distances, basically each turbine is just one tick mark regardless of their actual relative distance. This is a common way to display relative power along turbine rows, especially if some rows are short and others are long (e.g., Archer et al. 2018).*
*The WRF code calculates the power generated by each turbine, with both Jensen and Fitch. With Fitch, the wind speeds to calculate wind power are those at the grid centers, but, with Jensen, they are the actual Ui at each turbine in each grid cell.*

*We agree that the figures could be improved by removing the lines that connect the observations. However, the first author, who prepared the figures, left for another job in China and no longer has access to the computer cluster and is no longer in this field of work. Unless absolutely necessary, we would prefer not to change these figures. It would be a huge undertaking for us at this point.*

(8) One other concern is the intension to compare model results inside wind farms with measurements. The measurements are locally in strongly inhomogeneous conditions and the model instead is by design very diffusive in the horizontal direction with a true resolution far less than the model grid- size. Therefore, one should be very careful when trying to compare those two worlds.

*We added the following to the manuscript:*
*"We note that the power measurements at any wind farm are the result of local wind speeds at the turbines, often in strongly non-homogeneous conditions, while the model results by design are very diffusive in the horizontal direction, with a true resolution that is coarser than the grid size. As such, a direct comparison of the WRF and MPAS results against the observations reported in the next section should be interpreted with this limitation in mind."*

**Further comments**:

l.141: this link between the induction factor and thrust coefficient is only when momentum theory is applied.

> *The sentence was modified as follows:*
> *"and, after applying momentum theory, the induction factor a can be related to the thrust coefficient C_T by …"*

l.118: in the original definition the added TKE is the difference between the power extracted from the flow and the power converted in electricity. How, can the one quarter of the original value be theoretically justified?

> *There is not a theoretical reason for 0.25 as opposed to, say 0.3, or 0.1 (but it must be lower than 1 and positive). The value 0.25 simply fits the LES results best. The paper by Archer et al. (2021) actually discusses the limitations of this value extensively. For example: "We recognize that there is not one value that will work for all farms and all resolutions because the added TKE by a wind farm is a complex physical phenomenon that depends on more than just the thrust and power coefficients. However, the current formulation of the Fitch parameterization, especially after the bug fix proposed in section 2b, would dramatically overestimate the TKE added by the wind farm, and therefore even a general correction, such as the 25% factor proposed here, will give more realistic results than would no correction at all."*

> *But the point is that having no correction is theoretically and practically wrong because it implies no electromechanical losses between the turbine and the generator, whereas we know that energy is lost to turn the shaft, the generator, and the gears (if present) and because of other frictional losses. Also, most TKE in the far wake comes from shear generation in the wake, not from the tip vortices.*

l.18: it should be that wind turbine wakes *can* significantly decrease the wind farm power production. With high wind speeds and/or large turbine spacing the reduction at a downstream turbine can be low (or approach zero).

> *The sentence was modified as follows:*
> *"Wind turbine wakes can significantly decrease the wind farm power production (Archer et al., 2018)."*

l.39: it should be written that Volker et al. 2015 does *not* apply a TKE source term. Instead, it calculates a grid-cell averaged deceleration and the WRF model calculates the TKE due to the changed wind shear.

> *The sentence was modified as follows:*

*"a wind turbine is often parameterized as an elevated momentum sink (Volker et al. 2015) or as an elevated momentum sink and a source of turbulence within the vertical levels of the turbine rotor disk (Fitch et al., 2012; Abkar and Porté-Agel, 2015; Pan and Archer, 2018)."*

l.187: as stated in comments before, I would not agree that the first turbine row would face U∞. Also, what do you exactly mean with: there are no wakes at all inside the grid-cell? There is a grid-cell averaged deceleration, which is the consequence of the wakes inside the grid-cell.

*With the Fitch parameterization, each turbine in a grid cell produces the same power regardless of wind direction or layout because there are no internal wakes that would reduce the power production. Thus the Fitch parameterization does not consider any wake effects within a grid cell because it does not include a treatment for sub-grid wakes, thus it has no wakes in a grid cell. It generates a reduced wind speed as a result of the power extraction, not as a result of the wakes. Then the WRF dynamics advects that reduced wind speed downstream, which could be considered a resolved wake, but it is done by WRF and not by the Fitch parameterization.*

l.64: could you please clarify what do you mean with *ad-hoc* LES simulations.

*"Ad hoc" (from Latin "for this [purpose]") means done for a particular purpose as necessary. Here It means that, if you want a $\xi$ for farm A, you need to run LES of farm A first. If you want $\xi$ for farm B, you need to run LES of farm B. It is not a general correction factor.*

l.105: the three equations look a bit misleading, since there is only one source term listed in the model's deceleration (the same holds for the TKE). It would be clearer to introduce an additional source to the WRF deceleration (e.q. force/mass) and define the magnitude of that.

*We have followed the same notation and format of the Fitch parameterization paper and we have clearly stated that these 3 terms are "the momentum sink and TKE source terms induced by the turbines" only.*

l.185: what do you mean with: M4 is *generally* more accurate? and why?

*We present an extensive comparison of the performance of all methods, including M4, in subsequent sections. At this point of the paper, we simply give a quick "spoiler" that the new method is actually good. As shown later, it is the most accurate in all but perhaps 2 cases. We feel that using "generally" is sufficient here.*
*As for why M4 performs so well, please refer to our previous response to issue (III) on p. 3 of this response.*

**References**

Calmet I, and Mestayer PG, and van Eijk AMJ, and Herlédant O, A coastal bay summer breeze study, part 2: high-resolution numerical simulation of sea breeze local influences, Boundary-Layer Meteorol (2018) 167:27–51.

Kanth, A.L., and RaoS.V., Performance and Sensitivity Analysis of Very High-Resolution WRF-ARW Model Over Indian Region During 2011 Summer Monsoon Season, International Journal of Earth and Atmospheric Science, 2017, 4(04), 167-180.

Lean, H.W. and Clark, P.A., The effects of changing resolution on mesocale modeling of line convection and slantwise circulations in FASTEX IOP16, Quarterly Journal of the Royal Meteorological Society, 2003, 129 (592 PART A), 2255-2278.

Lynn, B. and Yair, Y., Prediction of lightning flash density with the WRF model, Advances in Geosciences, 2010, 23, 11—16.

---

## Author Comment (AC2)

**Response to Reviewer #2 - M. Paul van der Laan**

The authors develop a meso-scale wind farm parametrization based on the Fitch scheme coupled with an engineering wake model to represent the internal wake losses. In addition, a new wake superposition method is proposed. The predicted internal wake losses are validated with field measurements. Wind farm parametrizations cannot resolve internal wake losses and I think it is very interesting to overcome this by coupling an existing parametrization with an engineering wake model. The article has a focus on the internal wake losses; however, wind farm parametrizations in meso-scale models are mainly employed to estimate wind farm wake losses (for example wind farms situated in a wind farm cluster). Hence, I lack a validation of a wind farm wake produced by your proposed parametrization. You could for example look at the Horns Rev I wind farm wake as used by Volker et al. [1] or simply compare the wind farm wakes of the employed models (new and old) for the present idealized setup. More detailed comments are listed below; they need to be addressed before the article can be considered for publication in Wind Energy Science. Note that I cannot provide detailed comments on the numerical setup of the meso-scale models due to a lack of experience regarding this type of model. Also note that I have not yet looked at any other reviewer's comments before uploading this document because I want to provide an independent review.

> *The reviewer is absolutely correct that this work focuses on improving WRF and MPAS performance on internal wake losses by incorporating the Jensen wake model. Our study validates the models by comparing the predictions of internal wake losses versus field measurements (observational data) since investigating inter-farm interactions was not within the scope of this work. However, since inter-farm wake interactions originally initiate at the turbines, we believe that, if a model's performance in predicting internal wake effects improves, its inter-farm wake predictions would also improve.*

> *In addition, as the reviewer notes, wind farm parameterizations like the Fitch's one are often used to study inter-farm wakes. But it is important to point out that the inter-farm wake is not treated by the parameterization, but rather resolved directly by the WRF dynamics. As such, the wind-farm wake is going to be treated exactly the same whether the WRF is equipped with the Fitch or the Jensen parameterization.*

**Main comments**

1. Line 26: You mention that mainly two numerical approaches are employed to study wind turbine wakes: LES and WRF. However, there are a lot more numerical approaches to simulate wind turbine interaction, each model has its application area and purpose. Ranking steady-state wake models from low to high model fidelity one could think of: engineering wake models, simplified Reynolds-averaged Navier-Stokes (RANS) models as 2D RANS, parabolic RANS models [Iungo et al. (2018) [2]], linearized RANS (FUGA); and full 3D RANS with actuator disks (AD)[3]. Then we have transient wake models: Dynamic wake meandering model, unsteady RANS with actuator lines, LES-AD, LES with actuator lines (AL), WRFLES-AD and WRF with a wind farm

parametrization. Many of these model are described in G¨o¸cmen et al. (2016) [4], which you already refer to.

> *We meant to say "CFD," not just LES. All versions of RANS and URANS models, as well as LES, can be categorized as computational fluid dynamics (CFD) models. So, to fix this, we added the references and revised the paragraph as:*

> *"Mainly two numerical approaches are employed: computational fluid dynamics (CFD) and mesoscale modeling. Examples of CFD are Reynolds-averaged Navier-Stokes (RANS) models with various levels of sophistication, from 3D with actuator disks (van der Laan et al. (2015) to parabolic (Iungo et al. 2018), linearized, unsteady, or 2D (see Gocmen et al. (2016) for a review), and large-eddy simulation (LES) with actuator disks or lines."*

2. Line 74: Here you mention that the Jensen model works well for any wind farm layout based on the work of Archer et al. (2018) [5]. I do not agree with this strong statement since the performance of the model heavily relies on the wake superposition method and wake expansion coefficient. For example, the Jensen model does not work well for a tightly packed wind farm as Lillgrund and a below-rated inflow wind speed when the original quadratic wake superposition is applied (M2 method in the present work) [6], as you also show in the results section. I could not find the chosen superposition method in the work of Archer et al. (2018) [5], which makes is difficult to understand the results in that work. I think it would be better to write that the Jensen model could be calibrated to get the desired result for a given wind turbine type and wind farm layout.

> *We used method M2 in Archer et al. (2018) (it was not called M2 in that paper; please see equation 3 of that work on page 1189). In this paragraph, we mention two reasons for choosing Jensen: 1) it is possibly the most widely used model, and 2) it performs reasonably well regardless of the wind turbine layout or wind direction. We do not mean that Jensen is always the best model. Still, compared to other models, it appears more successful overall when applied to various wind farms using a wake expansion coefficient of 0.075 and 0.04 for onshore and offshore wind farms, respectively (Archer et al., 2018). According to our previous investigations, every wake loss model's performance strongly depends on the lateral and axial spacing between turbines. The more packed the wind farm is, the less accurate the models are. All models we have studied (and not just Jensen) show the worst performance in terms of absolute error and bias at packed farms (such as Lillgrund) and the best performance at the most widely spaced farms (such as Anholt). According to the extensive study by Archer et al. (2018), the Jensen model appeared to be one of the two models that showed a consistently strong performance among all types of wind farms (packed, moderately-spaced, and widely-spaced ones) and for all directions. Again, although in some cases it was not the best, it never appeared to be the worst either, and overall, it appeared to be one of the safest models one could choose.*

*With that said, we moderated the paragraph as:*

*"The Jensen model was selected for this parameterization … because it performs reasonably well regardless of the wind turbine layout or wind direction. …  While the Jensen model was not the best model all the time, it stood out for its consistently strong performance and for rarely ranking last for all directions and all farms."*

3. Section 2.2.2: It is interesting that you investigate different superposition methods. What is the physical meaning of method M4? I find it strange to superpose the absolute wind speed instead of the wake deficit. It means that you take a L2 norm of the wake wind speed normalized by the number of upstream turbine wakes, which is a form of averaging rather than a summation or superposition method. In other words, method M4 will always smooth out the wake effects between the largest and the smallest (upstream) wake wind speeds. There are also situations where the superposition method does not make sense. For example if one has a regular layout with a large lateral spacing (in y) but a small stream-wise spacing (in x), such that the turbines do not interact laterally, then a downstream wind turbine, will still experience reduced wake effects from the lateral turbines because you divide by the number of upstream turbines N.

*In the example of large lateral spacing and small streamwise spacing, M4 will not behave as described by the reviewer. The Jensen model predicts a finite wake laterally. If the lateral spacing is large, then the wake will not expand laterally enough to hit the turbines downstream, thus there will be no wind speed reduction (N=0, thus no wind speed reduction is applied). Basically, only the turbines that have a wide enough wake to inpact turbine i will be accounted for in the value of N. N is the number of turbines whose wakes impact turbine i, not just the number of upstream turbines or the number of turbines in a grid cell. This was stated at l. 163: "When multiple wakes from multiple turbines j (j=1...N) overlap at turbine i".*

*Having said this, the reviewer is correct that M4 will always give you basically an average of the wind speeds of all the wakes involved, which by definition will be higher than the lowest wind speed (and lower than the highest). This makes the most sense when the turbines are actually not aligned with the wind direction, which is the case for which this method was designed (multi-cell).*

*To understand this, let's look at the figure below, in which each turbine is in its own grid cell, and the wind directions are slightly different in each grid cell, as is the case in real simulations (also, the wind speeds are different, of course).*

[Figure]

*Turbine 4 is affected by 3 wakes, each of which causes its own wind speed (e.g., V4,1 is caused by turbine 1 and is a partial wake case, with some of the rotor affected by the wake and the rest unaffected, thus experiencing Vh,4). Suppose V4,1 = 6 m/s, V4,2 = 7 m/s, and V4,3 = 5 m/s. Note that these are the resulting wind speeds at turbine 4; they are not the undisturbed wind speeds at the upstream turbines. The resulting wind speed at turbine 4 will be 6.1 m/s, as a result of the mixing of the three individual wakes.*

*Why should the resulting wind speed be lower than 5 m/s? If that was the case, neither mass nor momentum nor energy would be conserved.*

*If the turbines are aligned, then this is less intuitive and may even appear to be incorrect. We are used to seeing the squared deficits from the Jensen's top hat shape overlapping and reducing the wind speed more and more with more overlapping wakes via the M2 method. This is actually not happening in the real world. We know that the wind power and wind speed plateau very quickly at the second or third turbine in a row due to partial wake recovery and entrainment compensating for the momentum extraction. If you look at the figures in our manuscript for directions 222° or 180° at Lillgrund or 339° and 183° at Anholt, you will see examples of observational data from the farms indicating that the third turbine generates more power than the second. With method M2, this trend cannot be reproduced; however, M4 appears to be successful in modeling such cases.*

*Method M2 (and M3) does not conserve energy or mass or momentum. Why should the squared deficits be conserved? They are a proxy for kinetic energy, but the mass is*

*missing, and the air is incompressible; thus, density cannot change, and it is impossible that kinetic energy sums up that way.*

*In addition, if we assume n equally distant aligned turbines, then the equation for M2 is basically a geometric series like this:*

$$\sum x^{2n}$$

*with |x|<1. Thus, as n increases, it converges to:*

$$\lim_{n\to\infty} \sum x^{2n} = \frac{1}{1-x} > 1$$

*which is greater than 1. This means that each additional wake deficit incrementally reduces wind speed and can even cause a zero wind speed or even negative. This is not physical and would never occur in reality. Method M4, however, does not suffer from this problem.*

*M4 does not conserve mass or momentum or energy either, but it will never result in a negative or zero wind speed and can be thought of as a way to account for entrainment and partial wake recovery. With M4, the resulting wind speed will be the average of the (partially recovered and therefore relatively high) wind speed from the farthest turbine and that (fresh and relatively low) of the nearest turbine, plus all the others in between. M4 effectively avoids the unrealistic continuous drop in wind speed as more and more turbines are aligned that other wake superposition methods suffer from. The fact that it outperforms other methods is also an indication that it is not incorrect.*

*We added the following additional sentence on p. 7:*
*"In a sense, M4 can be thought of as a way to indirectly account for partial wake recovery due to the added turbulence from multiple wakes and entrainment."*

4. How is the freestream wind speed defined/ calculated in your model? If it is the cell average, then this would mean that you would need to iterate over the Jensen model, since the cell average influences the internal wake model results, which influence the cell average. In case you use the freestream from the inflow, which is possible for an idealized case, how would you extend this to a realistic meso-scale setup where the inflow is transient?

*We use the cell average wind speed at the beginning of the time step as the free-stream wind speed. We had not thought of using an iterative process, but we suspect that it would further decrease the wind speed within the grid cell and may or may not converge to a constant. Hence, we apply the Jensen model only once.*

*We would like to bring up three important items to support our choice.*

*First, in the current Fitch parameterization, wind speed at every turbine within a grid cell (Ui) is always equal to the grid-cell wind speed (Uh), i.e., Ui=Uh all the time. This means that the current parameterization completely ignores the internal wake effects within the same grid cell. By contrast, the proposed Jensen-based parameterization uses the hub-height grid-cell wind speed (Uh) as the free-stream wind speed in the Jensen model to compute the wind speed that each turbine of that cell would experience.*

*Second, we acknowledge that Uh does not exactly follow the definition of free-stream wind speed. We struggled with it ourselves, but eventually we concluded that any alternative would be even more arbitrary and would potentially introduce more errors than using the grid-cell wind speed. For example, we could try to identify an upstream grid cell that could be considered a free-stream. But this grid cell would be wind direction-dependent and grid-size dependent. If you have a wind farm that is distributed among dozens of grid cells, each with its own wind direction and wind speed, which wind direction do you even pick to identify an upstream grid cell? If the wind farm is, say, 50 km long, does it really make sense to pick an upstream wind speed (and direction) that is 60 km away? What about the possible induction zone? We concluded that any other choice would be more arbitrary than just choosing Uh.*

*And third, a parameterization should be easy to implement and fast. It would be impractical to propose a parameterization that relies, for example, on idealized simulations without the farm, or on picking a grid cell outside of the wind farm somewhere. Choosing Uh is straightforward and easy to implement.*

*We changed the notation at line 160 to be consistent and replaced U(x)/U_inf with Ui/U_inf and added the above discussion to the manuscript as follows at p. 8:*

*"Alternative choices could be made for Uinf,i, such as the wind speed at a grid cell located at some distance upstream of the wind farm along one wind direction (among the varying wind directions simulated inside the farm), or the wind speed from offline simulations without the wind farm. However, any alternative would be even more arbitrary and would potentially introduce more errors than using simply the local grid-cell Uh. Another advantage is that Uh is straightforward to implement in the codes and fast to execute."*

5. How do you calculate the cell average wind speed from the Jensen model? Do you use the cell area only (and thus you would disregard the part of the wakes that extend beyond the cell area) or do you use an area that covers the entire wake of all the turbines within a cell?

*A cell's average wind speed, which is used as the free stream velocity for turbines within the cell, comes from WRF (Uh) and is not calculated via Jensen's model. Suppose Uh is the cell-average wind speed at a cell that contains turbine i. This average wind speed calculated at the previous time step is plugged into the Jensen model (Equation 6) as the*

*freestream wind speed to calculate U_ij, which is the velocity experienced by turbine i caused by an upstream turbine j, which is not necessarily in the same cell as turbine i. After wind speed or wind speed deficits caused by all turbines upstream turbine i are calculated, they are superimposed using a superposition method (methods M1 to M4). That yields total wind speed at turbine i and allows for calculating power production as well as updating u and v (using the exact same equations as those used by Fitch, Eq 1 and 2, with the only difference being a localized wind speed rather than the same one for all the turbines in the grid cell, Eq. 13-14), which would then be used for computing a new Uh at the next time step.*
*So, the average wind speed within a cell always comes from WRF, we did not calculate it (except for the vertical interpolation, like in Fitch's, to get the value at hub height).*

6. Equation (14): Should the left hand side be $\partial vk/\partial t$?

*Excellent catch. Thank you! We fixed it.*

7. Line 240: You propose to use a Gaussian filter with a standard deviation of 2∘, where did you base this value on?

*To our knowledge, there is no recommended value of the standard deviation to use in mesoscale simulations and we did not have time to verify the sensitivity of the results to the choice of the standard deviation for the Gaussian averaging. The value of 2° is loosely based on Gaumond et al. (2014), where they showed measurements with a standard deviation of 2.67° (their Figure 2). Since mesoscale models like WRF and MPAS are less variable than the observations and have already a smoothing effect, we wanted a value smaller than the standard deviation of observations and we rounded down to 2°. We added this to the limitations of our study on p. 34:*
*"In addition, we did not test the sensitivity of our results to the value of the standard deviation used for Gaussian averaging."*

8. Line 284: Why do you use a latitude of 50∘ ? Lillgrund and Anholt are located at 55.5∘ and 56.6∘, respectively.

*To be honest, we do not recall the particular reason. We probably mistakenly approximated the latitude of the two wind farms as about 50N. We do not think this mistake will have any significant impact on the results. For a previous study, we conducted LES using SOWFA to check the sensitivity of the simulated wakes to latitude and used the entire range from 0° to +/-90°. We computed the variation of the total power production of three inline turbines with an axial distance of 5D (see the following figures). We found that a difference of 5° around 50° has minimal impact on the wake deflection. Using linear interpolation, the total power production changes by only 0.014%.*

[Figure]

[Figure]

9. Section 3.1: Why are the measurements not filtered for neutral conditions? Without such a filter it is not fair compare the numerical results for neutral conditions with the measurements. If you prefer to keep the current measurements results you need to clearly state (in results section and conclusion) that the model validation is not entirely fair.

> *The reason we did not filter the data based on stability in this paper, even though we are painfully aware of its importance and have published several papers about it, is that the datasets we used did not include any information on stability. As such, we could not conduct any stability-dependent validation anyway, so we decided to stick with neutral stability for the simulations. We forgot to mention it and added it on p. 10:*

> *"No atmospheric stability information was available from the measurements at either wind farm."*

*and:*

*"Although we recognize that atmospheric stability may impact the evolution, shape, length, and duration of the wakes (Ghaisas et al., 2017; Xie and Archer, 2017), neither the Lillgrund nor the Anholt dataset included any measurements of atmospheric stability and therefore in this study we performed all the simulations under neutral conditions."*

*and in the Conclusions:*

*"Due to the lack of atmospheric stability information from the measurements, all the simulations were conducted under idealized neutral conditions."*

10. Section 3.2: It seems that you are modeling an idealized setup in the mesocale models. How do the steady-state inflow profiles of wind speed, wind direction and TKE/TI look like for both WRF and MPAS? You could plot these results in Fig. 11. What is the actual roughness length applied by the Charnock relation? What is the turbulence intensity at the investigated hub heights?

> *The Charnock relationship gives surface roughness from the surface friction u\*, thus it is a dynamic calculation. There is not one value of it, especially after the turbines are added. To access the values of $z_0$, we would need to rerun the simulations and select $z_0$ as an output variable.*
>
> *We agree that adding the undisturbed profiles of TKE and/or TI would be somewhat interesting. However, modifying the figures to add this information would be a very difficult task for us at this point. The first author, who was a postdoc working with us at the time of this research and who did all the figures, has moved on to a new life in China and we are having a hard time accessing and understanding his folders and files. Aside from the practical difficulty, we also would like to point out that we do not have any TKE or TI information from the measurements, thus we would not be able to gauge if the simulated profiles are realistic. This means that the additional profiles would not be terribly useful because they could not be verified. Also, we obtain results that are reassuringly similar to those from the Fitch parameterization, which suggests that our results are not incorrect or unrealistic. We could modify this figure by spending more time and resources on it if you find it absolutely necessary, but it would be a daunting task.*

11. Figures 3-8, measurements: What is the wind direction bin size used for the observations? In addition, you could consider to plot the error bars as the uncertainty of the mean (standard deviation dived by the square root of the number of samples).

*The wind direction bin size was 0.5 degrees. We added this information to the caption of Figure 3.*

*Regarding the second part of your comment, while dividing the standard deviation by the square root of the sample size may be a better error bar, unfortunately making this small change to the figures would be extremely difficult for us at this point. As mentioned earlier at #10, the first author, who was a postdoc working with us at the time of this research and who did all the figures, has moved on to a new life in China and we are having a hard time accessing and understanding his folders and files.*

12. I do not think it is fair to normalize the wind turbine power of Fitch with the power of the first row wind turbine, as Fitch is meant to model the average power of a number of wind turbines (or entire wind farm for the single cell case). You could overcome this by normalizing the wind turbine power of all the models and measurements with a hypothetical free-standing wind turbine power (using the freestream value at the wind farm location [without any wind turbines present] and the power curve. The measured freestream could be evaluated by using the power of a group of front row turbines and the power curve. A similar approach was performed in Hansen et al. (2015) [7].

*The definition of relative power is the ratio of the power of each turbine divided by that of the front-row turbine of each row/column. We have used this definition in this and all our other publications. It is not what the reviewer describes. Our treatment is fair because it is the same for Jensen and Fitch. Also, it actually helps visualize that, with the Fitch parameterization, all the turbines in the same grid cell are treated equally as front-row turbines, thus they truly should have a relative power of 1. When the wind farm covers multiple grid cells, the relative power from the Fitch parameterization is no longer 1.*

13. Line 478: You mention that the Jensen parametrization is insensitive to grid resolution: The Jensen parameterization tends to underestimate the power, regardless of the alignment or non-alignment conditions and regardless of the grid resolution. The consistent sign of the bias (negative) in column power output and the absence of sensitivity to the grid resolution or wind direction are all desirable properties. However, in order to show a grid independence you need to compare results of at least three different grid levels and show that the results converge with horizontal grid refinement. For example, you can add another grid level in Figs. 9 and 10, for example for 0.5 km or for 4 km. (The 4 km cell size might not be interesting for Lillgrund as this is the same at the single cell approach. (I am aware that grid-independence is challenging for WRF due to parametrizations that rely on large horizontal cells, but it might be possible for an idealized setup.)

*We have already run both models, WRF and MPAS, at three resolutions at each farm (24 km at Anholt and 4 km at Lillgrund for the single-cell cases, and 2 km and 1 km at both for the multi-cell cases). The 4-km resolution results are those called "single-cell" for*

*Lillgrund; the single-cell resolution for Anholt is 24 km because the Anholt farm is too big to be contained in a 4-km grid cell.*
*Also, we would like to point out that we have run two models for multiple wind directions (6 at Lillgrund and 8 at Anholt) and multiple wake overlapping methods, for a total of over 100 simulations.*
*We agree though that the term "absence of sensitivity" might be too strong. We modified the sentence as follows:*

*"The consistent sign of the bias (negative) in column power output and the minimal sensitivity to grid resolution or wind direction are all desirable properties."*

14. The cases considered in this article only look at high thrust coefficients, since a below rated inflow wind speed is used. The conclusion on the best superposition method might change if you had considered an above rated wind speed, where the thrust coefficients are smaller. This it because the performance of a super position method can be shown to be dependent on the thrust coefficient, see for example Machefaux et al. (2015) [8]. You could add an above-rated case or you could add a comment/discussion in the article.

*We added this discussion in the conclusion section as:*

*"A limitation of our study is that we only considered wind speeds that are below the rated wind speed, thus with a high thrust coefficient; when the wind speed is above the rated wind speed, the thrust coefficient decreases. Since the performance of a superposition method depends on the thrust coefficient (Machefaux et al. 2015), our results might be different in such cases."*

15. Line 370: You write: Both M3 and M4 reproduce well the feature of power output becoming steady after the fourth turbine in an alignment column (Fig. 4). I understand what you mean, but I think it is better to write about a balance instead of a steady-state. Here, I mean a balance between the momentum extracted by the wind turbines and the momentum transport into the wind farm from the boundary layer.

*We agree. The statement was revised as:*

*"Both M3 and M4 reproduce well the feature of power output remaining unchanged after the fourth turbine in an alignment column (Fig. 4), due to the balance between the momentum extracted by the turbines and that replenished from the boundary layer."*

16. Figure 11: I would normalize the wind speed by the freestream wind speed, $U_\infty$, and the TKE by $\sqrt{2/3}TKE/U_\infty$. The height can be normalized as $(z - z_H)/D$, with $z_H$ and $D$ as the hub height and rotor diameter, respectively. The same applies to Figs. 12 and 13, where the cell power production could be normalized by the total wind farm power.

*We agree that presenting data in a dimensionless format is beneficial when studying fundamental wake or turbulence behaviors. However, for practical applications like this one, we think that showing the actual wind speeds, the actual wind speed deficits, and the actual TKE is more valuable than the normalized versions.*

17. Line 496: You mention that a wind direction of 222◦ results in the largest wake losses for Lillgrund but this should 120 or 300◦ , where the wind turbine spacing is smallest.

*The paragraph was revised as:*
*"For these wind directions, the wake effects are neither strongest nor weakest, which allows for a representative comparison."*

18. Line 547: When referring to the Gaussian wake model, I think you need to refer to the original work of Bastankhah and Port´e-Agel (2014) [9].

*Done.*

**Minor comments**

1. Thanks for referring to my work. My last name is written with small letters for the first two words: van der Laan.

*Thank you for correcting us. We fixed this in the revised manuscript.*

---

## Referee Report (RR1)

**Review of *The Jensen wind farm parameterization for the WRF and MPAS models, R1* by Y. Ma et al.**

Reviewer: M. Paul van der Laan, DTU Wind Energy

July 29, 2022

The authors have mostly responded correctly to my comments and I accept the article to be published in WES. It is rather unfortunate that authors cannot change any figures because the first author is unavailable and this person was the only capable of making the plots. The proposed changes in figures from my side are minor so I think it should not be a big problem for the present article. However, I would recommend that more than one author can actually produce and modify the figures for future article submissions.

In my personal opinion, if an author is not willing to be part of the review process one could consider to remove this author from the author list and possibly add this person to the acknowledgments. This may seem harsh but all authors should be part of the review process as all authors can be held responsible for what is written in the article.

I have a few comments regarding the responses from the authors (that do not need to be addressed for present article):

**Main comments**

3. Thanks for clarifying M4, I was clearly confused about the parameter $N$. I follow your argument on M4 that it is beneficial to have a superposition method that does not lead to zero wind turbine power for those cases where we are not expecting below cut-in wind speeds. It is true that none of the superposition methods follow mass or momentum conservation but I am not yet convinced that M4 makes sense as a wake superposition method. I think a comparison the new superposition method with a full CFD model could be an interesting verification study.

4. You could consider an inversed 1D momentum method to estimate the freestream from a local cell-averaged wind speed similar to Abkar and Porté-Agel 2015 [1].

8. The effect of Coriolis forces on a small wind farm is indeed negligle as you have shown, but if you had investigated a larger wind farm, then the effect should be much stronger and you would probably have found larger impacts on wind turbine power, see for example Fig. 6.2 in [2]. I do agree that using 50 or 55 degrees for the latitude is probably not going to be critical for the results as the Coriolis parameter or surface Rossby number would only be 10% different and the effect of the surface Rossby number in the inflow is small for offshore surface Rossby numbers ( $10^9$), see for example Figs 6 and 7 from [3].

**References**

[1] M. Abkar and F. Porté-Agel, "A new wind-farm parameterization for large-scale atmospheric models," *Journal of Renewable and Sustainable Energy*, vol. 7, no. 1, p. 013121, 2015. [Online]. Available: https://doi.org/10.1063/1.4907600

[2] M. P. van der Laan, K. S. Hansen, N. N. Sørensen, and P.-E. Réthoré, "Predicting wind farm wake interaction with RANS: an investigation of the coriolis force," *Journal*

*of Physics: Conference Series*, vol. 625, p. 012026, jun 2015. [Online]. Available: https://doi.org/10.1088/1742-6596/625/1/012026

[3] M. P. van der Laan, M. Kelly, R. Floors, and A. Peña, "Rossby number similarity of an atmospheric rans model using limited-length-scale turbulence closures extended to unstable stratification," *Wind Energy Science*, vol. 5, no. 1, pp. 355–374, 2020. [Online]. Available: https://wes.copernicus.org/articles/5/355/2020/

---

## Author Response (AR2)

**Reply to Dr. Andrea Hahmann, Editor Reviewer**

Since the second referee was unavailable to review your answers to the revised manuscript, I have read and evaluated your answers and revised the manuscript myself.

I want to congratulate you on completing most of the revisions to the referee and my satisfaction. However, one of the points raised by referee Patrick Volker deserves some attention. I return to the point below.

*We are actually very thankful that you were willing to step in for Patrick. Please find our replies below.*

Q1. You made a nice explanatory figure in response to the reviews. Why not add it to the manuscript? It will be helpful to explain the various overlap options graphically.

*We added the figure to the manuscript as Figure 1 and discussed it as follows:*

*"Looking at Figure 1 as an example, Turbine 4 is affected by three wakes, each of which causes its own wind speed (e.g., $V_{4,1}$ is caused by Turbine 1 and is a partial wake case, with some of the rotor affected by the wake and the rest unaffected, thus experiencing $V_{4,0}$). The resulting wind speed at Turbine 4 is the result of the mixing of the three individual wind speeds. With M4 (Eq. 12), the resulting wind speed is basically the average of the (partially recovered and therefore relatively high) wind speed from the farthest turbine and that (fresh and relatively low) of the nearest turbine, plus all the others in between. In a sense, M4 can be thought of as a way to indirectly account for partial wake recovery due to the added turbulence from multiple wakes and entrainment.*

*On the other hand, method M4 may underestimate the deficit in the case of perfectly aligned turbines because it tends to dilute the wakes of the nearest turbines with the partially-recovered wakes of those further upstream. Even in the aligned case, however, M4 is effective because it avoids the unrealistic continuous drop in wind speed as more and more turbines are aligned that other wake superposition methods suffer from. While every wake superposition strategy introduces some level of under- or over-estimation, comparison with observational data, discussed in the next sections, indicates that M4 is generally the most accurate."*

Q2. I am missing a discussion section in the manuscript. L566-569 does not belong in the conclusions and could be included in this new discussion section.

*We note that Section 4 is titled "Results and discussion" because we did not want to just describe the results, but rather discuss them while we presented them. This is an intentional choice that we made because we believe that simply presenting results without a proper discussion is not only boring, but also less effective. It would be very*

*difficult at this point to separate the result presentation from the result discussion. Unless absolutely critical, we kindly ask to keep the current structure of the paper.*

*In an effort to ameliorate this issue, we renamed the Conclusions to "Conclusions, limitations, and future work".*

In addition, in my opinion, two critical points deserve attention.
(1) Wind farm parameterizations have been extensively used to study the possible farm-to-farm wakes (you cite a few in L46). [Many of the old studies are wrong because of the TKE advection bug in the WRF model that you helped to solve] These effects are much harder to evaluate in the parameterization because one needs data from two or more wind farms and/or many strategically located tall masts. But the important point is that improving the wind farm parameterization power production, as you have done in your manuscript, does not automatically improve the characterization of the wind farm wake, which is important for farm-to-farm studies. This point deserves to be mentioned in a discussion section.

*We added the following to the "Conclusions, limitations, and future work" section, L. 575:*

*"Another limitation is that we focused only on wind power production for validation. Wind farm parameterizations have been extensively used to study in-farm as well as farm-to-farm wakes. These wake effects on variables like temperature, humidity, turbulence, or heat fluxes are much harder to evaluate in a parameterization because multiple data sources are needed at and downstream of multiple wind farms, including strategically located tall masts and lidar measurements. Improving the power production alone, as we did in this study, does not automatically improve the characterization of the wind farm wakes."*

(2) The double counting of the wake effects is, in my option, real and should also be mentioned. Parameterizations, by definition, make many assumptions, and in your case, the wake of turbines in the upstream grid point could be accounted for twice. First, by the turbine wake overlap method and later when the WRF model advects the TKE and wind deficit downstream. By the way, the issue is not only in space but also in time. The wake from one turbine takes some time to travel to the next grid box. I don't mean that you should reformulate your scheme, but the point should be added to a discussion section.

*We added the following to Section 4.2, L. 437:*

*"when the Jensen parameterization is used, this effect is still present but it is added on to that of the sub-grid wakes. This causes a potential double-counting issue of the wake effects, which will be explored next by comparing results at various grid resolutions."*

*and the "Conclusions, limitations, and future work" at L. 580:*

*"Lastly, when the wind farm is partitioned over multiple grid cells (i.e., in the multi-cell cases) and the Jensen parametrization is used, there is the possibility of both resolved and sub-grid wakes being present simultaneously in the same grid cell, thus potentially double-counting some of the wake effects. By contrast, when WRF-Fitch is used, the resolved wake is the only wake effect that is accounted for, but it is generally too weak. When the Jensen parameterization is used, however, the resolved wake is still present, but it is in addition to the sub-grid wakes, which are generally stronger. Overall, we find that accounting for the (strong) sub-grid wakes with the Jensen parameterization, even in the presence of the inevitable (but small) resolved wake, gives more accurate results than relying on just the resolved wake. However, this issue needs to be investigated further, as discussed by Ma et al. (2022)."*

Q3. Adding your scheme to the WRF repository is a valuable goal. However, this could take time. I would strongly encourage you to make the source code for your scheme available to the community. Maybe the code could be available by request?

*We have submitted the request to add our scheme to the WRF repository to the WRF Physics Review Panel. Currently, this request is under review. While we wait, the code is also available by request. We added the following:*

*"While this request is under review, the code is available by request."*

In addition, I have a few minor points that should be corrected (lines refer to the ATC manuscript line numbers):

1. Please ensure that all units follow the WES guidelines of units formatted with negative exponents. Some of the units in the figure axis in the manuscript do not; also, in L518.

*The units in Figure 11 and in L518 are now written exponentially.*

2. I am old fashion, and I think the model WRF should be referred to as "the WRF model" not just "WRF."

*We agree that the acronym WRF does not contain the word "model" in it and therefore in many cases it is better to add it. However, we started adding it and it became a messy task very quickly. For example, many sentences were like this: "in WRF and MPAS" and they would have been replaced with "in the WRF model and in MPAS," because MPAS already contains "model" in the "M." This caused three additional words for one occurrence of WRF and did not add any clarity. If anything, it made the sentence awkward. There are some 64 instances of "WRF" in the paper, if we added "model" to each, we would add basically an entire paragraph with no real advantage.*

*We decided to state that we will use "WRF" instead of "WRF model" as follows at L. 74:*

*"… to two mesoscale models: the WRF model (referred to as just "WRF" or "the WRF" hereafter) and the Model for Prediction Across Scales (MPAS; Skamarock et al. 2012)."*

3. L17. It should be "..., and as wind turbine ROTORS expand in diameter..."

*Done.*

4. L18. "Wind turbine ... decrease wind farm power production." Concerning what? I suggest you revise the sentence.

*We clarified the sentence as follows:*

*"When a wind turbine wake hits a downstream turbine, it can cause a significant reduction in its power production; these wake losses negatively impact wind farm power production."*

5. L42. I think it will be better to cite a peer-reviewed paper than an unavailable conference abstract. I propose Volker et al (2017), https://doi.org/10.1088/1748-9326/aa5d86, since you are already citing one of P. Volker's manuscripts.

*Done.*

6. L44. The WRF abbreviation refers to Weather, Research and ForecastING.

*Revised.*

7. Figure 1. The wind turbines' locations are shown by black circles... BTW, what are the red arrows? It is not explained in the caption.

*The caption was updated as follows:*

*"The black arrows are the wind vectors at the grid cells; the red arrow is the wind vector at the grid cell of interest, replicated at the relevant upstream grid cells; and the green arrows are the average of the wind direction at the grid cells and that at the grid cell of interest."*

8. L264. "The Lillgrund wind farm is located in Sweden." -> "The Lillgrund wind farm is located in a narrow straight between Denmark and Sweden."

*Done.*

9. L354. Figure 3 shows THE relative...

*Revised here and similar expressions in the manuscript.*

10. L534. Maybe "Fig. 8a in Pan and Archer (2018)..."

*Done.*